# Using Knowledge Granularity Entropy to Measure Eco-Environmental Impacts of Land Cover Changes in ASEAN from 2001 to 2020

Weihua Liao

School of Public Policy and Management, Guangxi University, Nanning 530004, China; gisliaowh@gxu.edu.cn

**Abstract:** The eco-environment is the basis for the political, economic and social development of any nation or group of nations, such as the Association of Southeast Asian Nations (ASEAN). There is an inseparable association between land cover change (LCC) and eco-environmental change. The effects of the regional eco-environment are seen in the spatial and temporal variations in the eco-environment of different land cover types (LCTs). The Remote Sensing Ecological Index (RSEI), which is based on information granulation and spatial information granules, is used in this study to characterize the ecosystem. This issue is solved by breaking down the intricate ecosystem structure into straight-forward spatially granular computational units; this approach greatly reduces the complexity of spatial data computation. The main contributions of this study are as follows: (1) A model based on the concept of "granularity entropy" of the RSEI values of the different LCTs has been proposed by condensing and combining the spatial information granules. This model seeks to evaluate the stability or change of the eco-environment over time. (2) The main LCC factors for the decline in eco-environmental quality in ASEAN from 2001 to 2020 are the interval changes in eco-environmental indicator values caused by the decrease in forest area and the increase in grassland area; climate alteration is also a significant contributor to changes in regional eco-environmental quality.

**Keywords:** land cover change; eco-environmental change; knowledge granularity entropy; spatial aggregation and dispersion; ASEAN

## 1. Introduction

Land cover change (LCC) is the most intuitive manifestation of how human activities affect the Earth's surface system [1], which is a core area of global environmental change research. There is an unbreakable link between land cover and the eco-environment since land carries all human activities and plays a pivotal role in the ecosystem [2,3]. Material, energy and information exchanges and transformations among land cover types (LCTs) in the regional space work together to form a complex land ecosystem [4,5]. Land ecosystems are terrestrial socioecological systems in which humans and environmental systems interact through land use and emphasize the interdisciplinary fields of Land System Science [6–8]. An important factor influencing global ecological changes is represented by the eco-environmental patterns of LCC processes [9], which should be regarded as a momentous tool for investigating the harmonious coupling of human–land relations and sustainable development pathways [10]. The study of LCC and the environment will not only contribute to elucidating the process of LCC and how it influences the environment [11] but also provide a valuable scientific basis for sustainable land use and eco-environmental improvement, thus enriching the connotation of land systematics [12].

Land use and the eco-environment are closely and inseparably related. In recent years, scholars have analyzed the ecological effects of land from different perspectives and carried out numerous comprehensive studies. These studies include land ecological carrying capacity [13–15], land ecological security [16,17], land ecological restoration [18–20], land ecosystem services [21–23], and land ecological management [24,25]. Although land

ecology has many dynamic feedbacks and processes of change, it also exhibits some regularity [26]. Furthermore, the eco-environment characteristics of various LCTs vary [27]. Xu [28] proposed a remote sensing ecological index (RSEI), entirely based on remote sensing technology, to obtain ecological indicators for a comprehensive evaluation of the regional eco-environment. Since its proposal, many scholars have conducted rapid and accurate evaluation research of the regional eco-environment using RSEI [29,30].

Existing RSEI studies have mainly focused on assessing the quality of the regional eco-environment over a long period. However, ecological stability can also be impacted by modifications to the structure and layout of land use [31]. Land ecosystems are complex systems characterized by spatial and temporal changes, and new methods are required to characterize such changes. Information entropy can profoundly portray the relevant properties of complex systems [32]. System evolution is accompanied by changes in entropy [33]. Thus, the evolution of complex systems in land ecosystems can be described by changes in the entropy of the land cover system. As a transnational organization of different nations, the eco-environment shows a deteriorating trend and displays some volatility [28]. The LCC of ASEAN has already positively or negatively affected the eco-environment in multiple dimensions, including agriculture [34] and urban areas [35]. Therefore, new methods are needed to describe the eco-environment response caused by various LCTs.

In this study, we hypothesized that (1) each LCT has a distinct degree of change in the eco-environment over a time series that can be measured; and (2) there is some relationship between ASEAN land cover dynamics and eco-environmental degradation, allowing us to pinpoint which specific LCTs have caused this decline in eco-environmental quality. To test these hypotheses, we looked at eco-environmental change using remote sensing data (from MODIS) of land use between 2001 and 2020 in the ASEAN region. Our objectives were to (1) build a connection between land cover and the eco-environmental system by using the RSEI (greenness, wetness, heat and dryness) indicator to express the eco-environmental characteristics of land cover; (2) design an eco-environment indicator value change framework for land cover and the eco-environmental system in ASEAN and propose a method to calculate the change in knowledge granularity entropy (KGE); and (3) figure out the laws of eco-environmental change of different LCTs in ASEAN.

## 2. Materials and Data

### 2.1. Study Area

ASEAN is located in southeastern Asia, which is bordered by the Pacific Ocean in the east and the Indian Ocean in the west. As the "crossroads" between the Pacific Ocean and the Indian Ocean, Oceania and Asia, ASEAN occupies a crucial strategic position. ASEAN is divided into two major regions, the Malay Islands and Indochina Peninsula, consisting of 10 member countries of Philippines, Malaysia, Thailand, Indonesia, Singapore, Vietnam, Brunei, Cambodia, Myanmar and Laos (Figure 1). Natural resources abound in ASEAN, which have provided the necessary support for the sustainable development of the region and the world. Even while ASEAN only accounts for 3% of the total global land area, it is also rich in biodiversity [36]. However, in recent years, due to the rapid growth of the economy and demographic, the region's environment has been deteriorating in response to increasing consumption and waste of resources, which has hindered sustainable development. Despite having an abundance of natural resources, ASEAN, like other regions, faces a great challenge: how to maintain an appropriate balance between environmental sustainability and economic development.

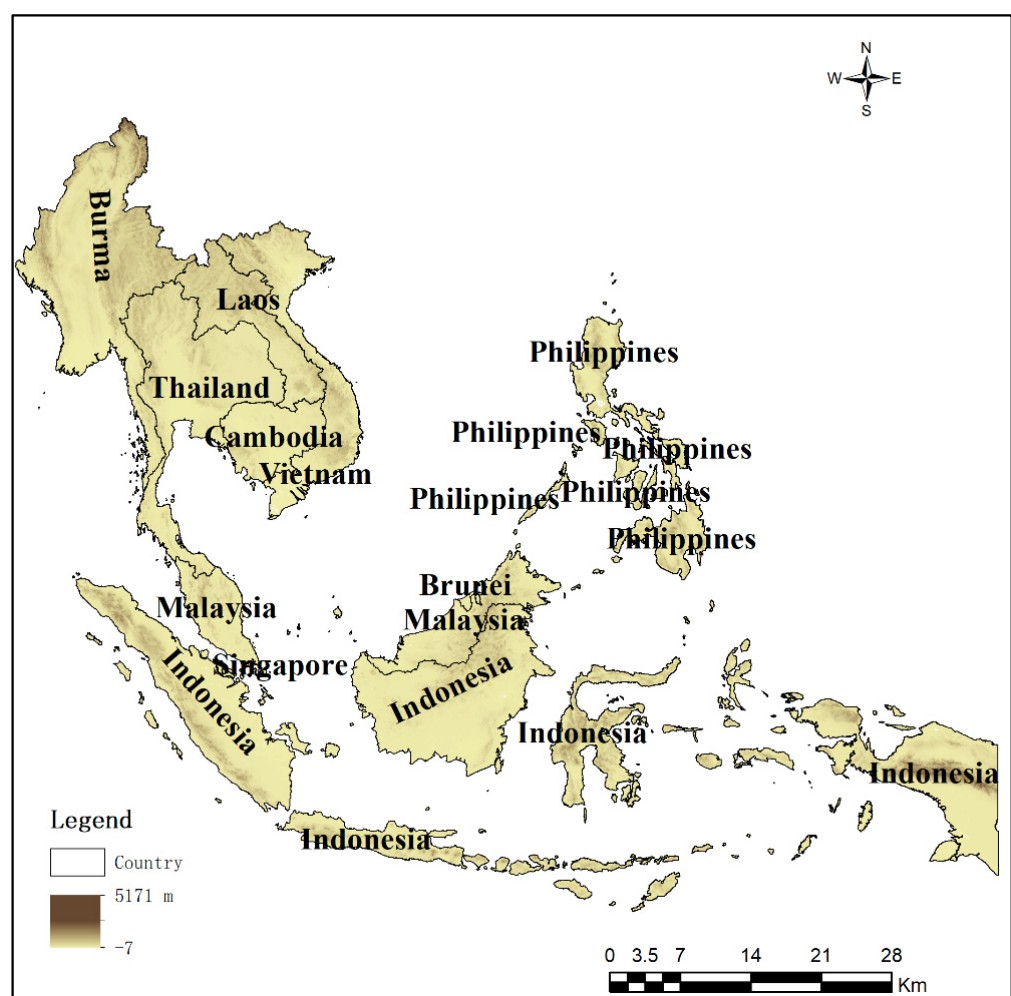

**Figure 1.** Location of ASEAN.

*2.2. Data Resources and Preprocessing*

The indicator data for greenness, wetness, and dryness used in this study were all acquired from the MOD09A1.006 product of NASA LP DAAC at the USGS EROS Center (https://doi.org/10.5067/MODIS/MOD09A1.006 (accessed on 20 January 2023)). Data on the land cover were sourced from the LC_Type1 band of the MCD12Q1 product of NASA LP DAAC at the USGS EROS Center (https://doi.org/10.5067/MODIS/MCD12Q1.006 (accessed on 20 January 2023)), and the classification criteria were adopted from the Annual International Geosphere-Biosphere Programme (IGBP) classification system, with 17 LCTs in total. The QC_Day band of the MOD11A2 V6 product, which has a spatial resolution of 1000 m and a period of every 8 days, provided the heat indicator data (https://doi.org/10.5067/MODIS/MOD11A2.006 (accessed on 20 January 2023)). The digital elevation model (DEM) was obtained from NASA JPL (https://doi.org/10.5067/MEaSUREs/NASADEM/NASADEM_HGT.001 (accessed on 20 January 2023)). National administrative boundary data were acquired from the Large Scale International Boundary (LSIB) dataset. All data were calculated in accordance with the calculation formula by using the GEE cloud computing platform (https://code.earthngine.google.com/ (accessed on 20 January 2023)).

## 3. Methods

*3.1. Indicators of the Eco-Environment*

This study makes use of the RSEI indicator system, which is based on remote sensing information technology. The RSEI can rapidly monitor and evaluate regional ecological

quality, and its indicator system involves four evaluation indicators, comprising the vegetation index, wetness component, surface temperature and soil index, which stand in for the four major ecological factors of greenness, wetness, heat and dryness. Among many natural factors reflecting ecological quality, these four elements are strongly tied to human survival. They are the most important indicators that permit humans to distinguish ecological conditions, so they are frequently used to assess ecosystems. The higher the value of the greenness and wetness indicators and the lower the value of the dryness indicator, the better the eco-environmental conditions. The heat indicator shows an optimum value, while higher or lower values signify deteriorating eco-environmental conditions. The formulas for calculating greenness, wetness, dryness and heat have appeared in a number of studies [29,37,38]. Greenness is conveyed by the normalized differential vegetation index (NDVI). The formula of NDVI is defined as follows:

$$NDVI = \frac{(Nir - Red)}{(Nir + Red)} \tag{1}$$

where Nir and Red are the reflectance of the band sur_refl_b02 and sur_refl_b01 band of MOD09A1.006, respectively.

The wetness formula is defined as follows:

$$\begin{aligned} WETNESS = {}&0.1147Red + 0.2489NIR1 + 0.2408Blue + 0.3132Green \\ &-0.3122Nir2 - 0.6416Swir1 - 0.5087Swir2 \end{aligned} \tag{2}$$

where NIR1, Blue, Red, Green, Swir1, and Swir2 are the reflectance of the sur_refl_b02, sur_refl_b03, sur_refl_b04, sur_refl_b05 band, sur_refl_b06 and sur_refl_b07 band of MOD09 A1.006, respectively.

The NDBI indicator represents dryness, composed of the index-based built-up index (IBI) and the bare soil index (BI). The following is a definition of the NDBI formulas:

$$BI = \frac{(Swir1 + Red) - (Nir + Blue)}{(Swir1 + Red) + (Nir + Blue)} \tag{3}$$

$$IBI = \frac{\frac{2Swir1}{Swir1 + Nir} - \left(\frac{Nir}{Nir + Red} + \frac{Green}{Green + Swir1}\right)}{\frac{2Swir1}{Swir1 + Nir} + \left(\frac{Nir}{Nir + Red} + \frac{Green}{Green + Swir1}\right)} \tag{4}$$

$$NDBI = (BI + IBI)/2 \tag{5}$$

The heat indicator is directly derived from the LST data of MODIS.

### 3.2. Spatial Information Granules

A spatially complex system, a system of spatial computation founded on continuous values, is created when spatial information is combined. Through spatial discretization, a complex superficial cognitive problem can be abstracted and simplified and then reconfigured to enrich the description of various spatiotemporal data [39]. Granular computing, which simulates the fundamental ability of people to deal with issues, is an intelligent processing paradigm whose center task is information granulation and computing based on information granulation [40]. The remote sensing image is a digital image segmented by spatial section, and the RSEI indicator pixel value is a continuous numerical space consisting of continuous values. In this study, the notion of spatial granules of remote sensing information is put forth. The spatial granule of remote sensing information is a spatial image formed by clustering the pixel values of a remote sensing indicator (RSI) image in compliance with some rules (such as interval, clustering, discretization, etc.) with the principle of spatial similarity due to spatial heterogeneity and continuity of the value for the indicator pixel value. By separating RSI pixels into RSI spatial images featured of discrete values according to different values, spatial granules of RSI information were

generated. Remote sensing spatial calculations can be simplified via the use of spatial granules of remote sensing information. For example, in Figure 2, the RSI image of a specific indicator (such as wetness) contains 12 pixels, and the indicator values are between –1 and 1. By dividing the values of pixels less than 0 into 1, 0–0.36 into 2, and greater than 0.36 into 3, the RSI image is discretized into an image of value codomains {1, 2, 3}. The RSI image information granules are converted into spatial granules of RSI information granule 1 with feature value 1, spatial granules of RSI information granule 2 with feature value 2 and spatial granules of RSI information granule 3 with feature value 3 relying on the reciprocity of the indicator feature values. Through information granulation of remote sensing images, the complicated continuous value calculation of remote sensing images can be reduced into a straightforward granular computing problem.

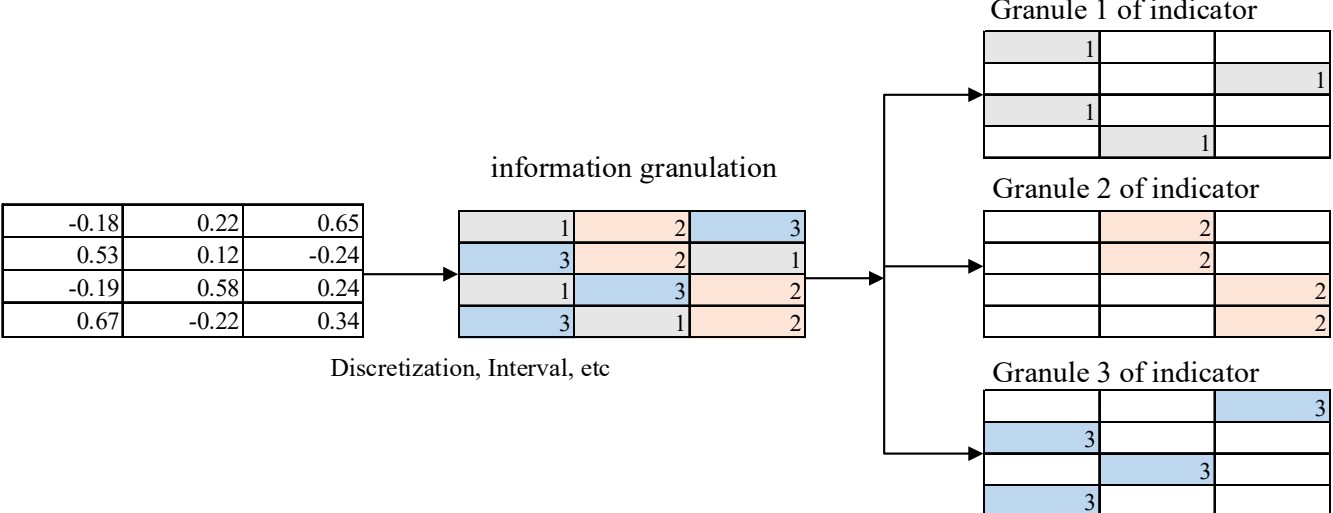

**Figure 2.** Process map of remote sensing information granulation.

Simple spatial granules of remote sensing information and composite spatial granules of remote sensing information constitute the two categories of spatial granules of remote sensing information. The simple spatial granules of remote sensing information classify the remote sensing images of an RSI hierarchically in accordance with the outcomes of spatial information granulation. RSEI indicators utilize rules to generate spatial granules of remote sensing information. Land cover images can produce simple spatial granules of remote sensing information depending on the classification results. As shown in Figure 3, the land image has 12 pixels and three LCTs, and three simple land spatial information granules are created on the grounds of the LCTs. Three simple spatial granules of land cover information come into being based on LCTs. The composite spatial granules of remote sensing information are the stratified division of remote sensing images after the overlay combination operation of two or more RSIs in the spatial image according to the consequences of spatial information granulation. As seen in Figure 4, greenness and wetness are images of 3 value codomains {1, 2, 3}. Therefore, the overlay operation to generate spatial images with 8 value domain types in the middle is the combined spatial granule of remote sensing information. In addition, 8 simple spatial granules of remote sensing information can be produced for the composite spatial granule of remote sensing information on the basis of 8 value codomains.

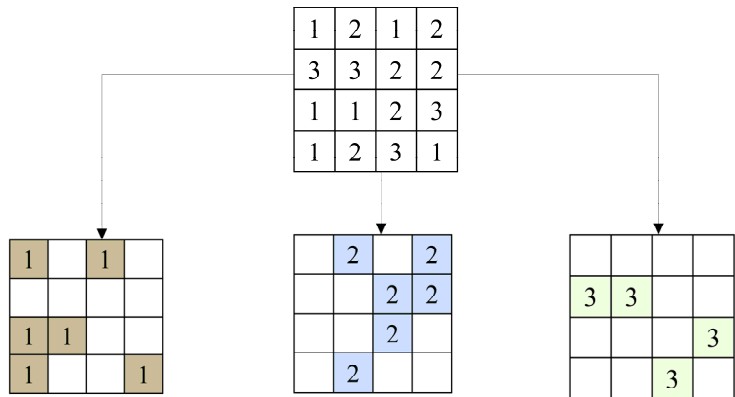

**Figure 3.** Simple spatial granules of remote sensing information.

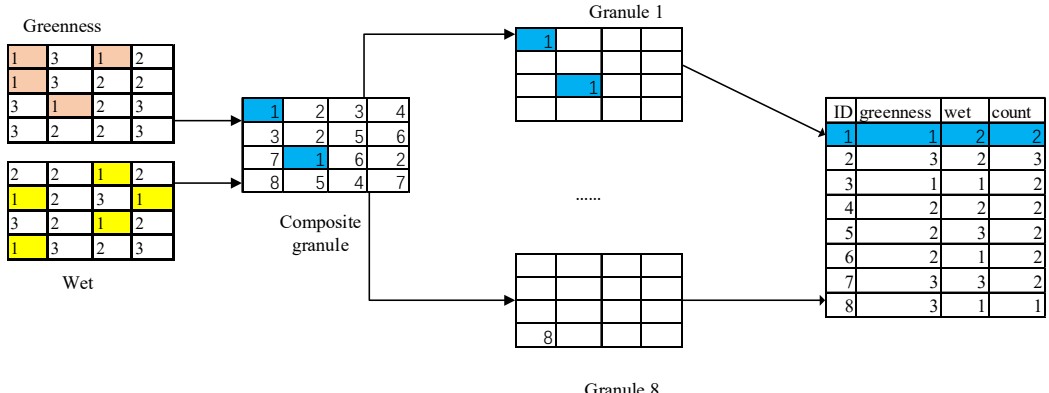

**Figure 4.** Composite spatial granules of remote sensing information.

### 3.3. Knowledge Granularity Entropy

The concept of entropy originates from classical thermodynamics and serves to measure how disorderly a system is. Shannon borrowed the term entropy and defined a metric to quantify the magnitude of randomness of a discrete random variable [41], referring to it as information entropy or random entropy. Information entropy is a measurement of system disorder, and the higher the entropy value, the higher the degree of disorder [42]. The measure of information entropy of remote sensing metrics is supplied by the following equation:

$$Er = -\sum_{i=1}^{n} p_i \ln(p_i) \tag{6}$$

where *Er* denotes the information entropy of RSI, n stands for all pixel numbers in the study area, and $p_i$ is the probability of different indicator values in all remote sensing image pixels. In the actual calculation of the eco-environment, the values of eco-environmental indicators are continuous. Since the numbers of the value of $p_i$ tend to be infinite, it is challenging to calculate *Er* directly [38]. Instead, the RSEI images are granulated into information spatial granule images with multiple levels of partitioning through spatial data information granulation, and this partition refinement can be known as the RSI knowledge granularity:

$$Gk(R) = \frac{\sum_{i=1}^{n} |X_i|^2}{|U|^2} \tag{7}$$

where *Gk* represents the knowledge granularity of the RSI, n indicates the number of spatial granules of RSI information for each indicator in the study area, $|U|$ denotes the total number of samples in the study area, and $|X_i|$ means the number of samples of each spatial granule of RSI information. Knowledge granularity is the measurement of the spatial distribution of the spatial information granules. The spatial information granules tend to

be more uniformly distributed the larger the measurement value. Based on the knowledge granularity of the RSI, the following KGE of the RSI is defined:

$$Er(R) = k \sum_{i=1}^{n} \frac{|X_i|}{|U|} log_2 |X_i| \tag{8}$$

where $Er(R)$ depicts the KGE of the RSI, and $k = 1/log_2|U|$. In an eco-environmental system, KGE is a measure of the spatial distribution disorder of spatial information granules [43]. The disorder is not a degradation or evolution of the eco-environment. There is a complementary relationship between KGE and knowledge granularity. The information granularity decreases with increasing KGE size. The smaller the KGE, the greater the information granularity. The eco-environment of any land cover type (LCT) would change due to the influence of external elements such as natural, social and economic factors, forming a dynamic process of spatial aggregation and spatial dispersion for indicators. Any object has a state distribution in space, which describes the organization and location of spatial objects in the objective world. When the RSI gradually tends from a state distribution to a centralized distribution of some indicator information granules, a process of spatial aggregation for the RSI is developed. In contrast, when the RSI gradually moves from a state distribution to the equilibrium distribution of some indicator information granules, a process of spatial dispersion of the RSI is formed. For instance, in Figure 5, a specific land cover greenness indicator has three levels of 1, 2 and 3, and the corresponding values are 5, 7 and 4, respectively. In addition, its KGE value of RSI is 0.613 after computation. Under the influence of the outside world, changing to the left, the numbers of the three levels of greenness indicators become 5, 6 and 5, the KGE value of RSI is 0.605, and the three levels of greenness indicators are inclined to be more balanced in spatial distribution, which is a process called spatial dispersion. If another external influence shifts to the right, the numbers of the three levels of the greenness indicator move to 1, 14 and 1, and the KGE of the RSI is calculated to be 0.833. The three levels of greenness indicators have a tendency to be spatially concentrated in level 2, and the greenness spatial information granules prefer to be concentrated and distributed, which is a process of spatial aggregation.

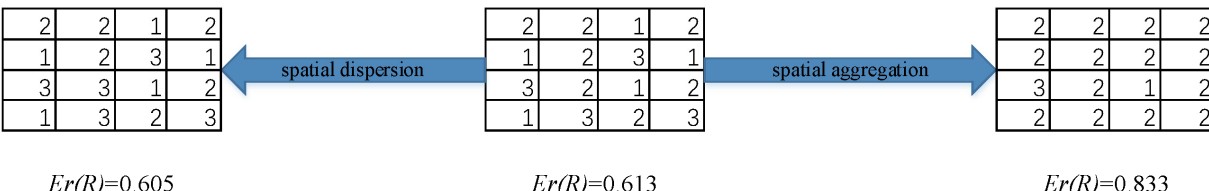

**Figure 5.** Spatial aggregation and dispersion process in land cover.

The KGE of RSI for land cover in long time series is a dynamic change process, and the extent of eco-environmental alteration in land cover is a cumulative value of the KGE of RSIs, which is defined as follows:

$$En(R) = \sum_{i=2}^{n} (Er(R)_i - Er(R)_{(i-1)}) \tag{9}$$

where $En(R)$ is the eco-environment entropy change degree of the LCT, n is the year's evaluated number, and $Er(R)_i$ stands for the KGE of the RSI from the second year onward. If $En(R)$ is $> 0$, the eco-environmental change of land cover is a spatial aggregation process, and the eco-environment is likely to be both concentrated and spread. If $En(R)$ is $< 0$, then the modification in the eco-environment of land cover is a spatial dispersion process, and the eco-environment possesses an inclination to be distributed in a balanced way.

*3.4. Analysis Framework*

Based on the available data, computing platforms and the concepts proposed above, we presented the remote sensing ecological change analysis framework for LCC in Figure 6. The framework is divided into three main parts.

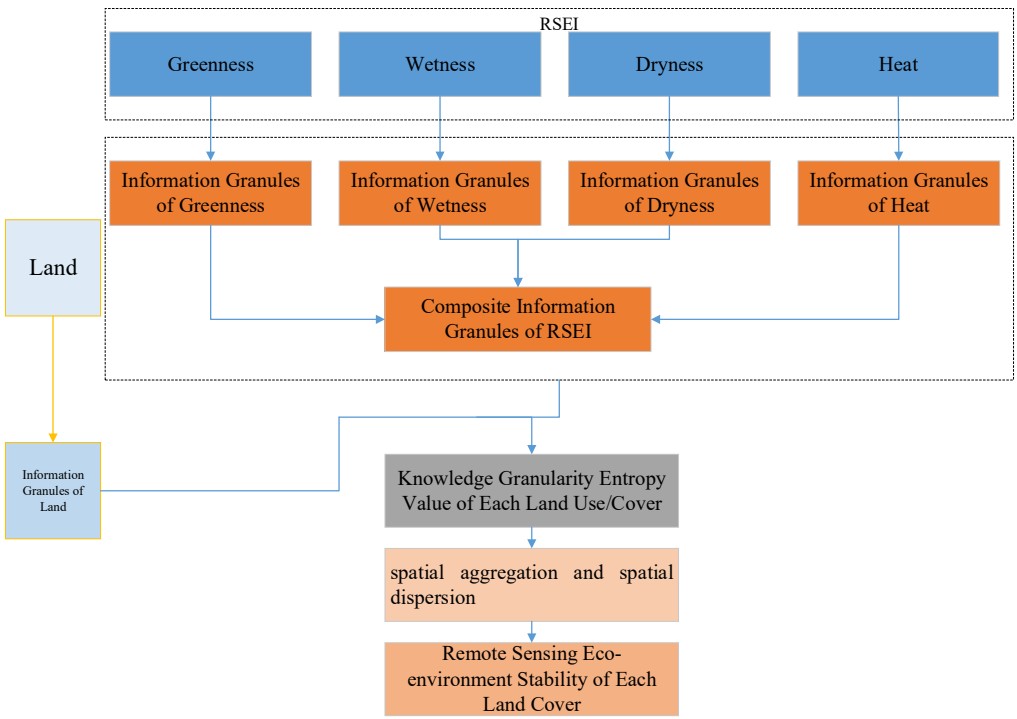

**Figure 6.** Analysis framework of land cover and eco-environmental change.

(1) Four images of RSEI indicators for each period of each year of ASEAN from 2001 to 2020 are collected on the GEE platform, and each indicator is used to figure out the average value of each year via the aggregation data technique. First, each variable is separated into 20 eco-environment simple spatial information granules of that parameter by an equal distance. Then the four metrics are integrated to generate RSEI composite spatial granules.

(2) Remote sensing images of land cover in the ASEAN region are gathered for each year from 2001 to 2020 and split into 17 spatial granules of land cover information that correspond to 17 LCTs.

(3) The degree of variation in the entropy of the eco-environment is calculated for each LCT in each year. Next, the spatial aggregation or spatial dispersion process is ascertained for the KGE of each land cover from 2001 to 2020, and the causes of the change in the eco-environment of LCTs in the ASEAN region are also examined.

## 4. Results

Evergreen broadleaf forests predominate in the LCTs in ASEAN, which account for approximately 50% of the entire region each year. The LCTs in ASEAN primarily consist of forest, grassland and agricultural landscapes, and the top five LCTs in terms of percentage are evergreen broadleaf forests, woody savannas, cropland savannas, and cropland/natural vegetation mosaics, which constitute nearly 85% of the area of the entire region; these landscapes laid the foundation for the regional eco-environment. The area of ASEAN evergreen broadleaf forests, which make up about 6% of the land area in the whole region, significantly decreased from 2001 to 2020 as a result of the influence of social and economic activities, and evergreen broadleaf forests degenerated into other land types, turning them into the main factor of regional ecological environment deterioration. A more pronounced degradation trend is also visible in mixed forests and cropland/natural vegetation mosaics.

The area of woody savannas in ASEAN has increased greatly by around 5%, and the area of deciduous broadleaf forests has also risen notably, as shown in Figure 7. All LCTs in ASEAN prove a very obvious spatial dispersion phenomenon (Figure 8), and the land types have a tendency to aggregate to a uniform distribution, but the KGE value is high, and the whole region still assembles on a few major LCTs. The regional eco-environment is built on the entropy change caused by the structural evolution of LCTs.

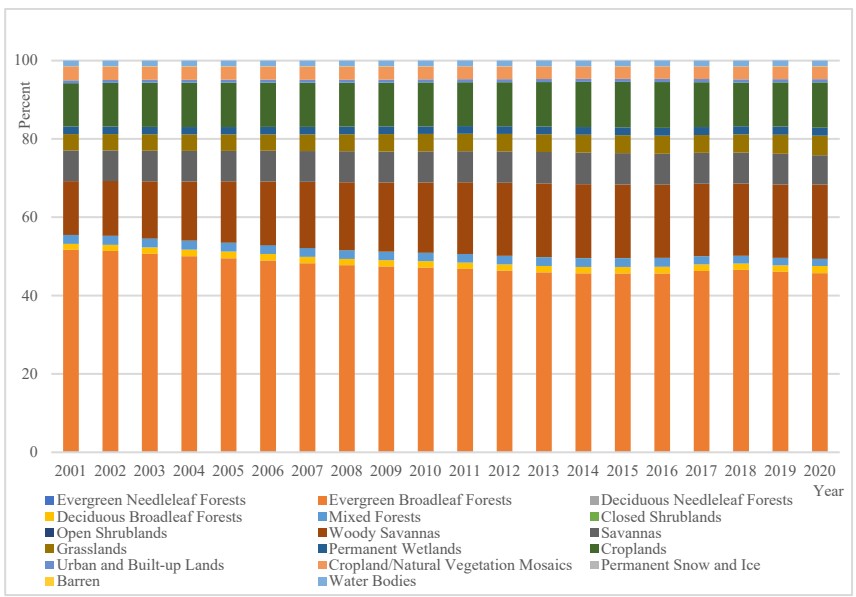

**Figure 7.** Map of the proportion of land cover in ASEAN from 2001 to 2020.

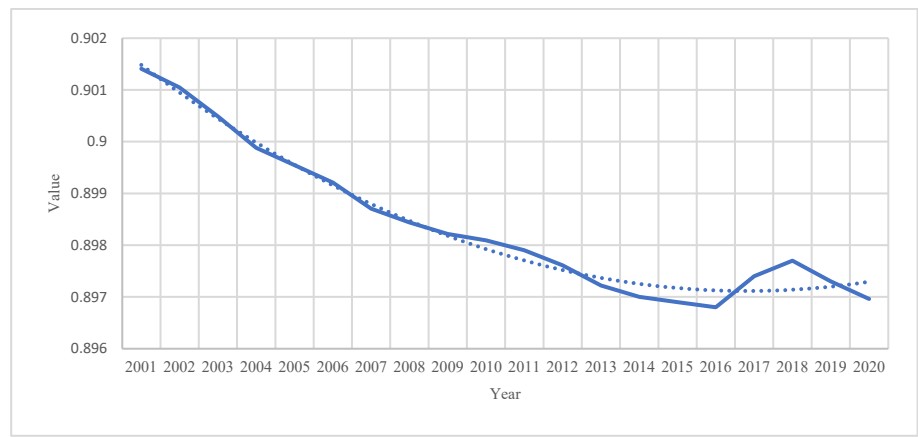

**Figure 8.** Map of knowledge granularity entropy value for land cover in ASEAN from 2001 to 2020.

Permanent snow and ice, croplands, and cropland/natural vegetation mosaics were the top three LCTs in ASEAN over the study period, with average KGE values of the greenness index. Because of snow and ice cover, permanent snow and ice had a single ground cover. The average KGE value of two agricultural landscape LCTs, namely, croplands and cropland/natural vegetation mosaics, was high, which indicated that the long-term changes in crop planting types in the research area were not noteworthy, and there seemed to be concentrated distributions of some crops. Deciduous needleleaf forests, open shrublands, and closed shrublands made up the bottom three LCTs with average KGE values of greenness indicator in ASEAN. Due to the diversity of vegetation cover types, shrublands demonstrated the uniform distribution of spatial information granules at all greenness levels greater than level 7; the KGE values of greenness signals of various LCTs are displayed in Figure 9. The top three LCTs in terms of eco-environment entropy change degree were

deciduous needleleaf forests, deciduous broadleaf forests, and grasslands, and it was a spatial aggregation process. Deciduous broadleaf forests and grasslands both aggregated toward the interval of the higher values of the greenness indicator. The bottom three LCTs in terms of eco-environment entropy change degree values were open shrublands, barren, and permanent snow and ice, and it was a process of spatial dispersion. Other LCTs with obvious spatial dispersion and spatial aggregation processes were smaller in the area; any minor shift in the interval of greenness indicator values will cause great KGE value fluctuations. Figure 10 illustrates the eco-environment entropy change levels of greenness indicators for various LCTs.

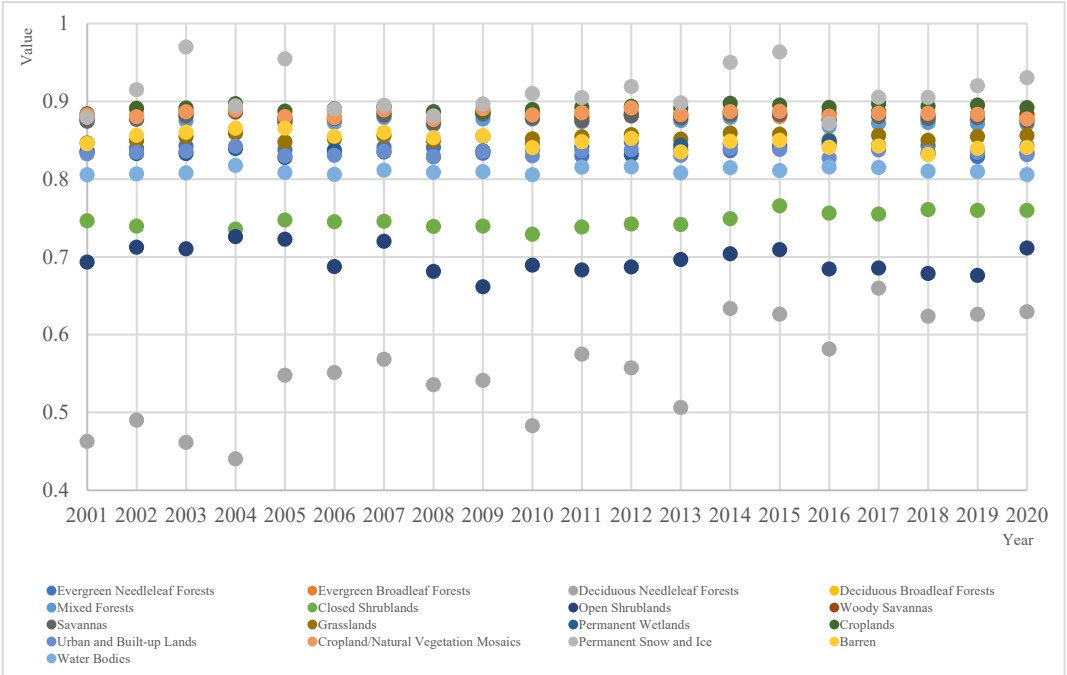

**Figure 9.** Map of knowledge granularity entropy value of greenness indicator for land cover in ASEAN from 2001 to 2020.

During the study period, the top three LCTs with average KGE values of wetness indicators in ASEAN were evergreen needleleaf forests, deciduous broadleaf forests, and woody savannas. Evergreen needleleaf forests with wetness indicators were mainly focused on the middle value interval, which protected soil water content and enhanced the eco-environment. The wetness of broadleaf forests was mostly in the middle value interval, which had a protective effect on soil wetness content, while the wetness of woody savannas was primarily in the low-value interval, which had a low soil wetness content and had a suppressive impact on the regional eco-environment. The last three LCTs with average KGE values were deciduous needleleaf forests, open shrublands, and closed shrublands. The wetness intervals of open and closed shrublands were largely concentrated in some high-value intervals, which had a certain beneficial impact on soil wetness content. Figure 11 displays the KGE values for various LCTs' wetness index. The lowest three LCTs in terms of eco-environment entropy change degree value for wetness were permanent snow and ice, open shrublands, and deciduous needleleaf forests. Open shrublands had a tendency to cluster in the high wetness indicator interval, and the areas of the other two LCTs were comparatively small. Barren, woody savannas and closed shrublands were the bottom three LCTs in terms of eco-environment entropy change degree value of wetness. Barren and closed shrublands were evenly distributed in several high-value intervals, and woody savannas tended to be uniformly spread in low-value intervals. The eco-environment entropy variability of the wetness indicator for various LCTs is shown in Figure 10.

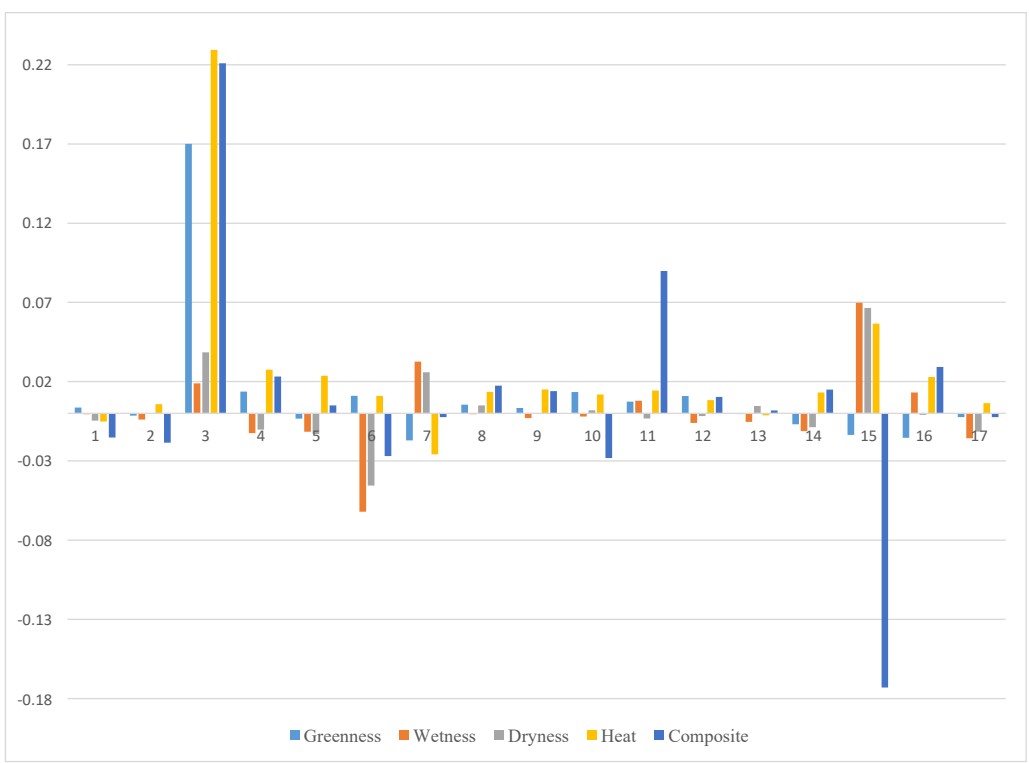

1: Evergreen Needleleaf Forests 2: Evergreen Broadleaf Forests 3: Deciduous Needleleaf Forests 4: Deciduous Broadleaf Forests 5: Mixed Forests

6: Closed Shrublands 7: Open Shrublands 8: Woody Savannas 9: Savannas 10: Grasslands 11: Permanent Wetlands 12: Croplands

13: Urban and Built-up Lands 14: Cropland/Natural Vegetation Mosaics 15: Permanent Snow and Ice 16: Barren 17: Water Bodies

**Figure 10.** Change degree of eco-environmental entropy for land cover in ASEAN countries from 2001 to 2020.

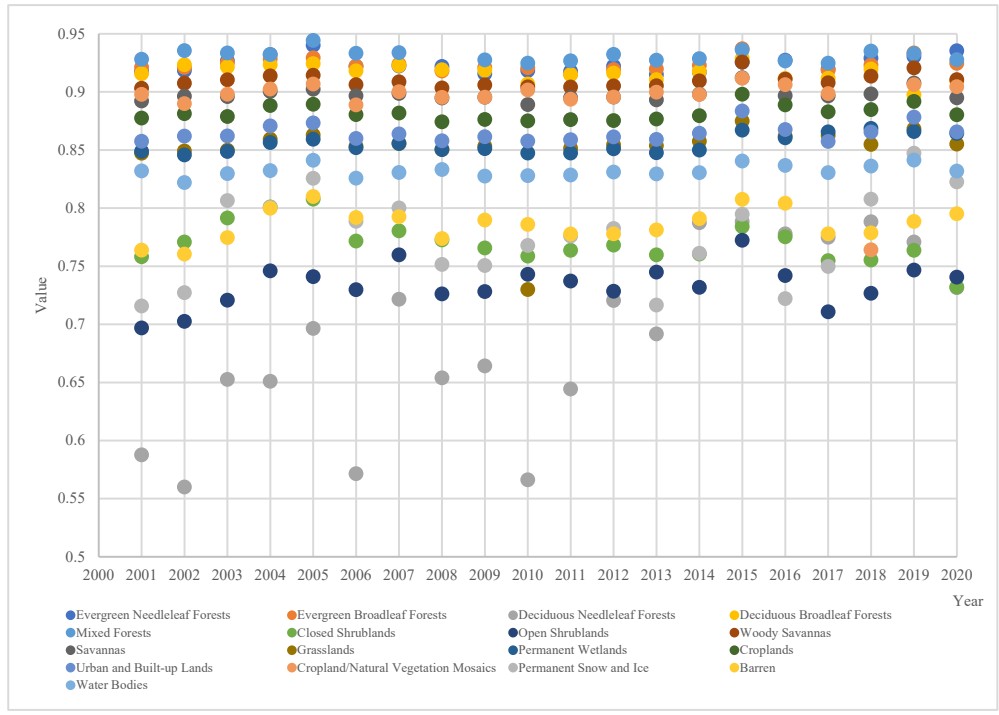

**Figure 11.** Map of knowledge granularity entropy value of wetness indicator for land cover in ASEAN from 2001 to 2020.

Mixed forests, evergreen needleleaf forests, and evergreen broadleaf forests were the top three LCTs with average KGE values of dryness in ASEAN over the course of the study. The dryness of mixed forests was mainly centered in the high-value interval, and the surface of this LCT was getting drier. Evergreen needleleaf forest and evergreen broadleaf forest dryness were especially concentrated in the low-value interval. As a result, their land surface turned progressively wetter, and they had an advantageous effect on ecological improvement. The lowest three LCTs with average KGE values of dryness indicators in ASEAN were deciduous needleleaf forests, open shrublands, and barren areas. The intervals of dryness indices of these three LCTs mostly occurred in low-value intervals, and the ground surface was close to the natural ground surface. Figure 12 indicates the KGE values of various LCTs' dryness indicators. The highest three LCTs in terms of eco-environment entropy change degree values for dryness were permanent snow and ice, open deciduous needleleaf forests, and open shrublands, all of which clustered around the high-value interval of dryness; the dryness of the land surface continually increased, which contributed to ecological and chemical environment deterioration. Closed shrublands, mixed forests, and water bodies were the bottom three LCTs in terms of eco-environment entropy change degree value for dryness. Closed shrublands and mixed forests seemed to be equally split among several high-value intervals, and water bodies were likely to be evenly distributed between high- and low-value intervals. The remote sensing ecological entropy variability of the dryness for each LCT is represented in Figure 10.

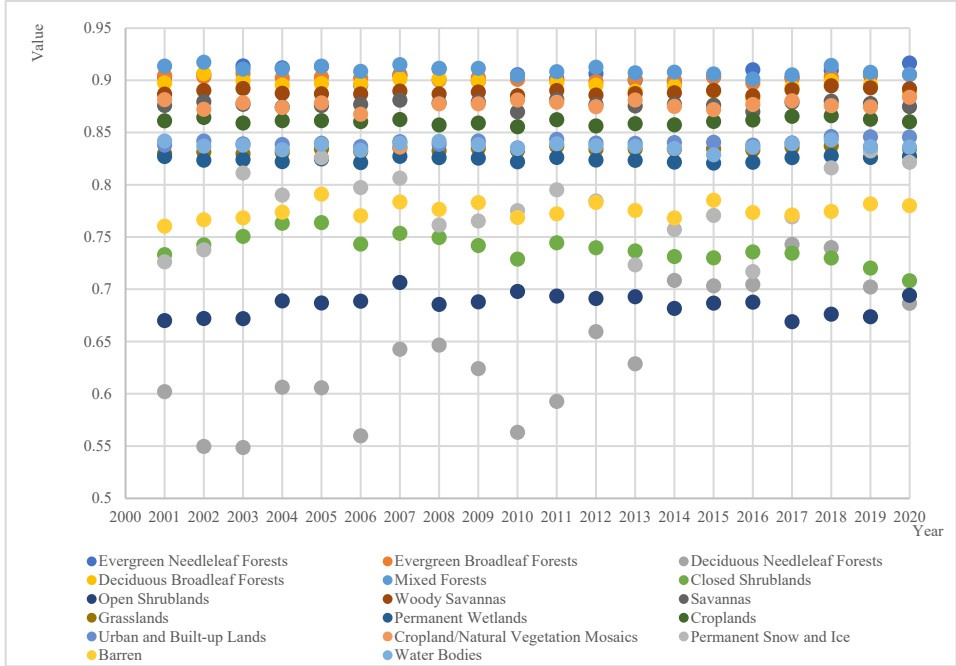

**Figure 12.** Map of knowledge granularity entropy value of dryness indicator for land cover in ASEAN from 2001 to 2020.

The leading three LCTs with average KGE values for heat in ASEAN during the research time were water bodies, mixed forests, and woody savannas. Water bodies' heat values were primarily in the medium interval, and mixed forests and woody savannas' heat indicator values were mainly in the low interval. Deciduous needleleaf forests, open shrublands, permanent snow and ice, were the next three LCTs with average KGE values for heat in ASEAN. The heat intervals of open shrublands and permanent snow and ice were essentially uniformly distributed in some low-value intervals. The KGE values of heat for various LCTs are demonstrated in Figure 13. The most prominent three LCTs in terms of eco-environment entropy change degree values for heat were deciduous needleleaf forests, permanent snow and ice, and deciduous broadleaf forests. Permanent snow and

ice and deciduous broadleaf forests have a propensity of moving and clustering from the low-value interval of the heat indicator to the high-value interval, and glaciers, snow cover and deciduous broadleaf forests were all damaged by global warming. Open shrublands, evergreen needleleaf forests and urban and built-up lands were the bottom three LCTs in terms of eco-environment entropy change degree value for heat. Urban and built-up lands and open shrublands tended to be fairly spread in several high-value intervals, and temperature increase was positively accelerated by urbanization, whereas regional ecological improvement was restrained. Figure 10 depicts the eco-environment entropy variability of heat for each LCT.

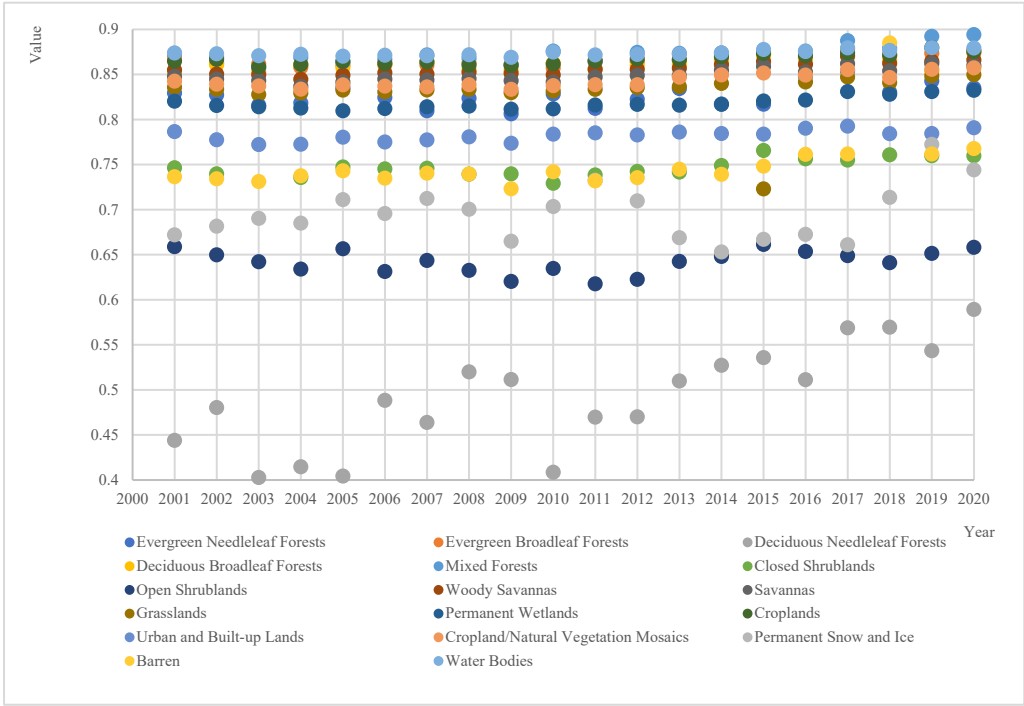

**Figure 13.** Map of knowledge granularity entropy value of the heat indicator for land cover in ASEAN from 2001 to 2020.

The top three LCTs of average KGE values of the RSEI composite indicator in ASEAN over the study period were permanent wetlands, croplands, and deciduous broadleaf forests. The high KGE value of the composite indicators indicated that these three LCTs tended to be focused on some intervals of the composite indicators. Deciduous needleleaf forests, open shrublands, and closed shrublands made up the bottom three LCTs of average KGE values of the RSEI composite indicator in ASEAN. These LCTs had low KGE values of the composite indicators, showing that these three LCTs appeared to be distributed in some intervals of the composite index. The KGE values of the composite indicators for various LCTs are displayed in Figure 14. The highest three LCTs in terms of eco-environment entropy change degree values were savannas, deciduous needleleaf forests, and barren. The composite granules of the composite indicators of these three LCTs assembled several combinations of indicators, which had a strong influence on eco-environmental change. The bottom three LCTs in terms of eco-environment entropy change degree values of RSEI composite indicators were permanent snow and ice, open shrublands, and grasslands. The collective granules of combined indicators of these three LCTs were headed to be evenly distributed among certain combinations of indicators. Figure 10 demonstrates the degree of eco-environment entropy change for the combined indicators of various LCTs.

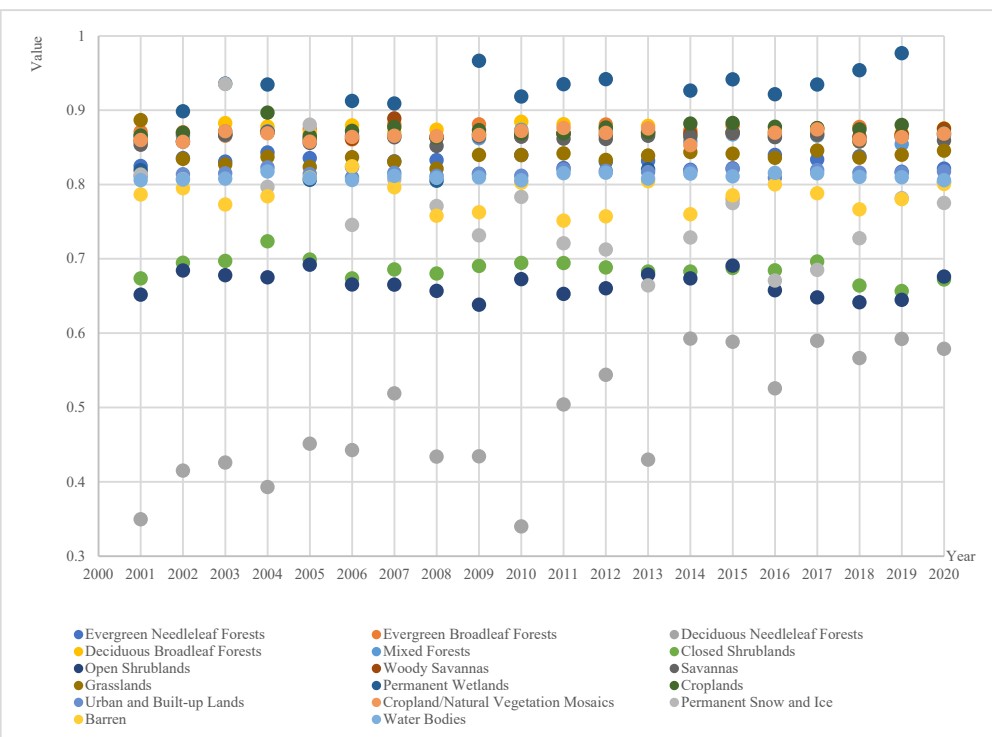

**Figure 14.** Map of knowledge granularity entropy value of composite spatial granules for land cover in ASEAN from 2001 to 2020.

## 5. Discussion

### 5.1. Structural Change

Regional ecosystem quality is a dynamic process of change, and remote sensing ecosystem quality in the ASEAN region has revealed a declining trend over the past 20 years [39]. The eco-environment is the sum of various natural forces or actions that affect human life and production activities [44]. The Earth's surface is a complex system composed of several LCTs, and the eco-environment is also a complicated system constructed out of various indicators [45]. Systems are interconnected in many ways, such as feedback and response [46], and structural changes in one system have an impact on another related system [47]. With KGE based on spatial RSI information granules, complex land and remote sensing ecosystem systems may be decomposed into structured granules. They may also employ general structured problem-solving methods to think about and solve spatial computing problems. There are 17 types of land cover in ASEAN, and the two LCTs with relatively large eco-environment entropy change degree values are deciduous needleleaf forests and permanent snow and ice, which share a smaller area. Their drastic changes in the eco-environment will significantly influence the local eco-environment while having little effect on the whole region. The degraded area of evergreen broadleaf forests in the ASEAN region accounted for approximately 6% of the whole regional area during the study period. In the ASEAN region, evergreen broadleaf forests are a typical zonal forest vegetation type and a crucial LCT for maintaining the regional ecological balance. Evergreen broadleaf forests, with regard to greenness, wetness, and composite granules, are spatial dispersion processes, shifting from the high-value interval to the low-value interval of indicators. In addition, the structural alterations in two aspects of dryness and heat are weak spatial aggregation processes, with an incentive to equally distribute to the high dryness interval and high heat interval. Woody savanna areas comprise about 5% of the whole area. The structural adjustments in other indicators except wetness showed a weak spatial aggregation process, with greenness and wetness transferring from high values to the median value interval. Dryness and heat are moving from low values to the median interval. In addition, the primary drivers of the decrease in eco-environmental quality in the

ASEAN region are the structural changes in the eco-environment caused by the variation in crop cultivation structure, urban construction, and other human economic activities.

### 5.2. Hierarchical Change

An ecosystem is a sophisticated system with a hierarchical structure, and numerous earlier studies have considered the ecosystem a holistic system that relies on continuous values [48–50]. Remote sensing ecosystem indicators for various LCTs are characterized by a hierarchy in space and time. Knowledge granularity is applied to measure the amount of information available for analyzing and processing data in different hierarchically structured spaces [51,52]. The multilevel and multi-granular cognition model satisfies the requirements of human cognition and problem-solving, and multilevel cognitive computing is an efficient solution to address the contradiction between the cognitive model of land and remote sensing ecology and the computing model of spatial data. The same land cover and eco-environmental problems from different levels can be observed and analyzed through spatial information granulation computing and spatial information granules. As shown in Figure 15, when one indication of greenness is used to evaluate the eco-environmental change of land cover, the computation is a relatively simple operation and a straightforward cognitive calculation. When adding the wetness indicator, a composite information spatial granule is formed; this granule can more accurately determine how the eco-environmental change of land cover has changed. As a result, the KGE reduces when an increasing number of indicators are included. Therefore, the model is more capable of capturing the spatial calculation problem. For example, evergreen broadleaf forests in the study area consisted of 13 spatial information granules with a KGE of 0.8773 for the greenness indicator in 2016, 102 composite information spatial granules with a KGE of 0.8155 for greenness and wetness, 1036 composite information spatial granules with a KGE of 0.6914 for greenness, wetness and heat, and the number of composite information spatial granules with a KGE of 0.6526 for greenness, wetness, heat and dryness is 2459.

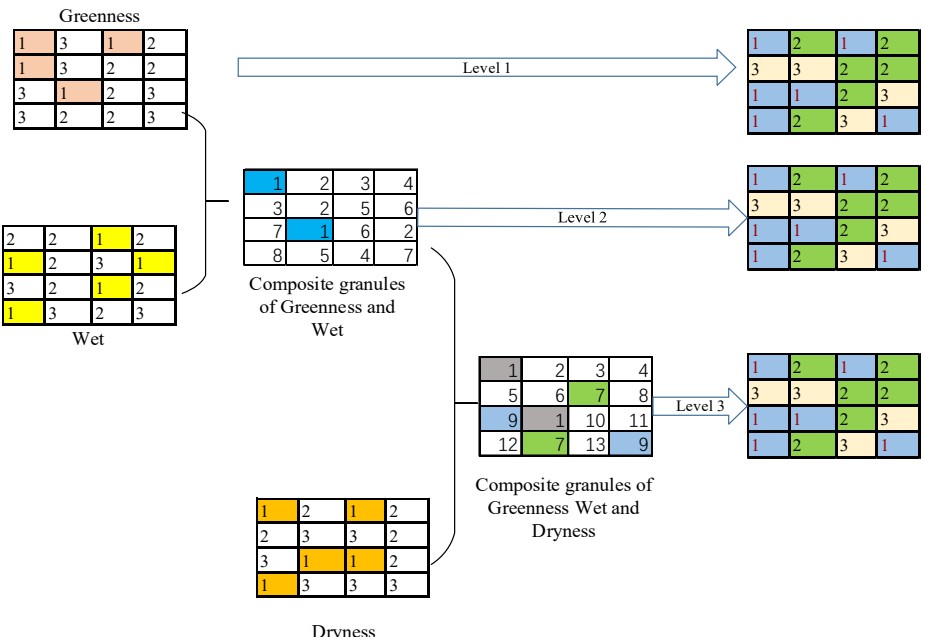

**Figure 15.** Hierarchical change diagram of granule combinations with different indicators.

### 5.3. Comparison with RSEI

The RSEI reflects the ecological and environmental quality of a region, and to a certain extent, it can also mirror the relationship between humans, social activities and environmental quality, which has an enormous impact on sustainable development. The RSEI-based eco-environmental quality evaluation of a region can supply decision assistance

for the continued growth of the region [53–55]. RSEI examines regional remote sensing ecological and environmental quality changes across time and space as a whole. Still, it is a weight calculation that depends on an indicator system, and the correlation between indicators is the basis of evaluation studies [56,57]. The majority of the existing RSEI studies do not calculate the correlation between several indicators. In this study, the RSEI was adopted to express the ecosystem of LCTs. Eco-environment entropy change degree value only conveyed the stability change of the ecosystem but failed to take into account the rank of ecosystem quality, which is unrelated to correlation. Comparing the LCT changes of ASEAN for two years in 2001 and 2020, 6.728% of evergreen broadleaf forests in the whole area of the region moved to woody savannas, 0.633% to savannas, 0.806% to grasslands, and 1.353% to woody savannas. The main factor of eco-environmental quality degradation in ASEAN is the conversion of forest to grassland. Figure 16 represents the major spatial changes in land cover. The KGE built around spatial information granulation computing and spatial information granules can discover the connection between LCC and interval changes in RSEI indicators and thus measure the causes of RSEI changes in the region from another perspective.

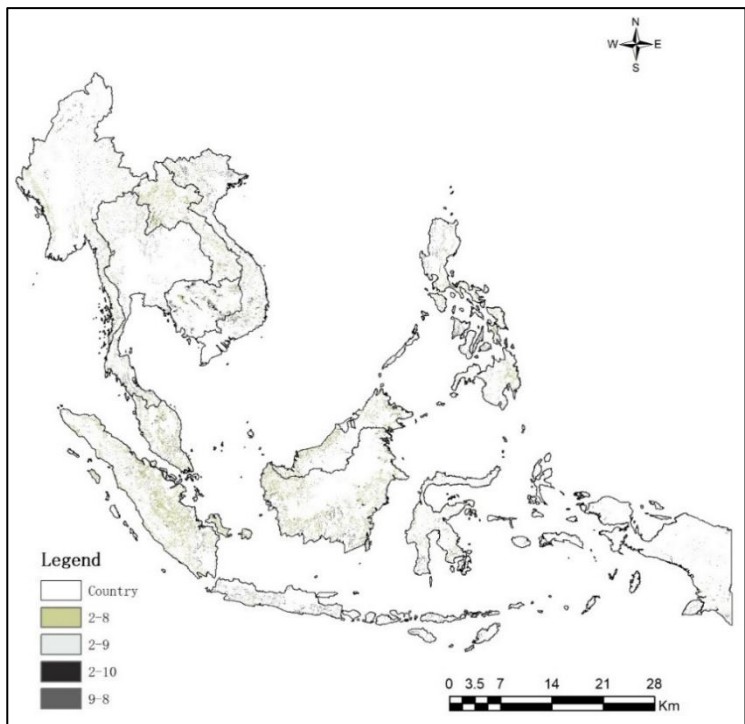

**Figure 16.** Spatial changes in the main land cover types in 2001 and 2020. The value in the legend represents the land cover change from 2001 to 2020; e.g., "2–8" represents the type of land cover change from 2 in 2001 to 8 in 2020.

### 5.4. Implications for Policy-Making

The transnational and indivisible nature of ASEAN environmental problems indicates that these problems cannot be resolved by any one country acting alone but instead require concerted cooperation among countries within and outside the region on all aspects affecting the regional environment, such as biodiversity reduction, seagrass and mangrove conservation, and tropical rainforest protection. ASEAN must leave behind the traditional state-centered governance model, strengthen effective interregional coordination and cooperation, continue to deepen the construction of free trade zones in other cooperative organizations, including China, and take examples from each other's experience in environmental governance. In order to adequately fund environmental cooperation, ASEAN should boost its economy and establish a permanent regional environmental management

agency. ASEAN should recognize that climate change has emerged as one of the biggest threats to long-term regional stability. It ought to motivate stakeholders to participate in the action plan and increase efforts to decarbonize the public and private sectors to collaborate with other regions of the world to simultaneously limit global warming to 1.5 °C. Land ecological degradation can be prevented through collaborative legislation, real-time monitoring of all components of natural resources by using high-tech tools such as remote sensing, prohibiting tree cutting, returning farmland to forests to prevent soil erosion, and planting trees and grasses.

*5.5. Limitations and Future Research*

The change in the eco-environment of different LCTs can be measured by the KGE based on spatial information granulation computing and spatial information granules. Still, this alteration is not a criterion for judging the change in RSE quality. The spatial aggregation and spatial dispersion phenomenon of the eco-environment of LCTs is the drastic degree of change of this type of eco-environment, and it is necessary to qualitatively determine the evolution direction of eco-environmental quality by referring to the interval change of eco-environmental indicators. Even if the study has granulated each indicator information into 20 spatial information granules, the question of whether equidistance division is the best approach for computing spatial information granulation can continue to be investigated. Future work can use the spatial association of spatial information granules to identify the interconnection between LCC and the high and low zones of RSEI factors; this information can then be utilized to more precisely determine the changes in eco-environmental quality brought on by changes in land cover changes and the quantitative metric system of various interconnections between RSEI factors and LCTs.

**6. Conclusions**

To better analyze and address the systemic problems of the land eco-environment, our research proposed a spatial data structured solution method founded on spatial information granulation computing and spatial information granules. This method can abstractly divide the complex eco-environmental problem into several simple problems. The spatial aggregation and spatial dispersion process based on KGE can measure the direction of eco-environmental changes of different LCTs to determine the connection between LCC and eco-environmental quality changes. The spatial aggregation process of deciduous needleleaf forests, permanent snow and ice, and the spatial dispersion process of closed shrublands are all readily apparent. In other indices, other LCTs exhibit weak spatial aggregation and spatial dispersion processes.

Most likely, the decrease in forest area, particularly the notable decline in evergreen broadleaf forests, is the main reason for LCT change, causing the drop in remotely sensed ecological quality in ASEAN from 2001 to 2020. The area of evergreen broadleaf forests has shrunk by about 6%, and its remote sensing ecological entropy change degree values in greenness, wetness, dryness, heat, and composite indicators are −0.0014, −0.0039, 0.0001, 0.0058, and −0.0184, respectively, during 2001–2020. The transformation from evergreen broadleaf forests and other forest LCTs to woody savannas is the secondary cause of the loss in remotely sensed ecosystem quality. In the meantime, the primary factor contributing to the decrease in RSEI quality in the ASEAN region is the interval variation in RSEI indicators for each LCT resulting from climate change.

**Funding:** This research was funded by Guangxi Natural Science Foundation grant number 2020GXNS-FAA297176 and National Natural Science Foundation of China grant number U21A2022,42071393.

**Institutional Review Board Statement:** Not applicable.

**Informed Consent Statement:** Not applicable.

**Data Availability Statement:** Not applicable.

**Conflicts of Interest:** The author declare no conflict of interest.

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
