# Peer review of "Using Knowledge Granularity Entropy to Measure Eco-Environmental Impacts of Land Cover Changes in ASEAN from 2001 to 2020"

_sustainability, doi:10.3390/su15119067_

Round 1

Reviewer 1 Report

This study is overall interesting. Some revision suggestions are as below.

1. It should include the author's detail information, such as city, postcode and email address

2. Some key findings should be presented with data in the abstract section.

3. The Figures and tables should stand alone, and some abbreviations should indicate the full name.

4. In the introduction, the author must explain the innovation points and research gap clearly. And some latest literature should be provided.

5. The Figures in this study should have vertical axis names and horizontal axis names. Such as Figure 12, Figure 13, etc.

6. The conclusion section is too simple. Please present the research results step by step and illustrate with data. At the same time, please made the conclusion.

Author Response

Thank you for your valuable review comments. Your comments have greatly helped me improve my manuscript and research level. I have made revision to the manuscript item by item based on the opinions of the reviewers.

This study is overall interesting. Some revision suggestions are as below.

  1. It should include the author's detail information, such as city, postcode and email address

Response: Thank you for reminding me. As it was the first draft submitted at the time, I did not include detailed author information in the first draft. Now in the revised manuscript, I have improved the author’s information.

  1. Some key findings should be presented with data in the abstract section.

Response: Like question 6, the determined and refined data has no direct significance for the explanation of the results. It is more of a directional method for determining the eco- environment. Of course, there are still shortcomings in this method, and I have also explained them in the manuscript.

  1. The Figures and tables should stand alone, and some abbreviations should indicate the full name.

Response: Thank you for your suggestion. I have already changed all the abbreviations of the Figure titles to uppercase in the revised manuscript.

  1. In the introduction, the author must explain the innovation points and research gap clearly. And some latest literature should be provided.

Response: There are many ways to write Introduction, and I know that some SSCI papers prefer to separately list research gaps, shortcomings, motivations, and other parts in the introduction section. I firmly believe that the writing of the introduction section of this manuscript is standardized and can reflect the research background, significance, and various research requirements of the study. Of course, based on your proposal, I have refined this section to better highlight the purpose of the research and understand the gap between existing research and this study.

This manuscript cites 13 papers published in 2022, as well as 3 papers on sustainability, but does not cite papers published this year

  1. The Figures in this study should have vertical axis names and horizontal axis names. Such as Figure 12, Figure 13, etc.

Response: Based on the modification opinions of you and other review experts, I have also improved the horizontal, vertical, and legend of the graphs.

  1. The conclusion section is too simple. Please present the research results step by step and illustrate with data. At the same time, please made the conclusion.

Response: The conclusion is general and a summary of the research contribution. The results of this study are too many data to be listed one by one. Its main contribution is to provide a way to judge the change in eco-environment quality from the change of entropy value of various ecological indicators on land cover. In the research limitation part, I also said that a fully quantitative method has not been found to judge this change. Of course, I also enriched the conclusion section appropriately

Reviewer 2 Report

Motivation should be more detailed in the introduction.

Contributions to the literature and knowledge should be placed in the introduction, and reinforced and abstract should be rewritten.

Legend in figure 9 and onwards is mandatory.

In all figures where legends are announced to check figure 7, please provide the legend as figure notes. They should be self-contained and easily understandable just by looking to them. 9, 10, 11....etc.

Figure 1 is really necessary? A Table with the main elements would not be enough?

Keep consistency along the text; namely titles in bold and others no...

Correct minor mistakes in the EN.

The quality of the figures provided should be improved.

Author Response

Thank you for your valuable review comments. Your comments have greatly helped me improve my manuscript and research level. I have made revision to the manuscript item by item based on the opinions of the reviewers.

Motivation should be more detailed in the introduction.

Response: Perhaps due to differences in writing habits, this manuscript provides a detailed description of the research motivation in the last paragraph of the introduction. Of course, based on your suggestions, the revised manuscript has refined the research motivation.

Contributions to the literature and knowledge should be placed in the introduction, and reinforced and abstract should be rewritten.

Response: There are many ways to write Introduction, and I know that some SSCI papers prefer to separately list research gaps, shortcomings, motivations, and other parts in the introduction section. I firmly believe that the writing of the introduction section of this manuscript is standardized and can reflect the research background, significance, and various research requirements of study. Of course, based on your proposal, I have refined this section to better highlight the purpose of the research and understand the gap between existing research and this study.

Legend in figure 9 and onwards is mandatory.

Response:  Based on the modification opinions of you and other review experts, I have also improved the horizontal, vertical, and legend of the graphs.

In all figures where legends are announced to check figure 7, please provide the legend as figure notes. They should be self-contained and easily understandable just by looking to them. 9, 10, 11....etc.

Response:  My initial consideration for the construction of this legend was to save image space. Based on your and other reviewers' opinions, I have redrawn these diagrams according to the specifications.

Figure 1 is really necessary? A Table with the main elements would not be enough?

Response: When studying an area, providing a location map is a common process in many studies. A descriptive table with detailed data can achieve the same effect, but I still respect the traditional use of area maps.

Keep consistency along the text; namely titles in bold and others no...

Response: Thank you for your feedback. The editor has made preliminary adjustments to my submission, but some may not have been properly handled. I have rechecked the layout and formatting.

Correct minor mistakes in the EN.

Response: I have checked the entire text to avoid spelling, grammar, and other errors in English as much as possible.

The quality of the figures provided should be improved.

Response: I have redrawn many images, both for standardization and for image quality. Inserting images in Word will reduce the resolution of the images. If the publisher is willing, I can provide high-quality original images to ensure the quality of the images.

Reviewer 3 Report

I had high expectations for this manuscript; the subject matter immediately struck me as particularly interesting and the approach seemed interesting and innovative. Unfortunately, all I got out of it was a deep disappointment and I am really sad about that. It seems like a wasted opportunity.

In my opinion, there are too many flaws in this manuscript and it should therefore be rejected.

I made several attempts to read it, but I had to make a considerable effort to get to the end and understand its content, its purpose, the methodology used, the results obtained and their relevance.

Firstly, the English language used is often inappropriate for a scientific article. The text is wordy, unnecessarily complicated, redundant in many terms, imprecise, vague, uncertain, sometimes even incorrect. A radical overhaul of the English language would therefore be absolutely necessary, taking into account the need to use correct and specific terminology, to better organize sentences so that they are fully comprehensible, and to avoid too many unnecessary turns of phrase.

Terms are often used without being defined in a rigorous way, as they should appear in a scientific paper. To give you just a few examples, what is meant by 'eco-environmental quality'? The concept of “quality” does not mean anything without a precise definition. Moreover, what is meant by “land use structure and layout”? What is meant by 'structural change'? What does the following expression mean? “The eco-environment shows a deteriorating trend and exhibit some volatility”. What is meant by 'deteriorated or degraded area'? What is meant by “volatility” in this specific context? This is very imprecise and causes confusion. It also increases the frustration of reading. Too often terms are used that should have a precise meaning, but no specific definition is given. This is a far cry from the scientific rigour required. For this reason, it was particularly difficult to read the text and fully understand its meaning.

Even more serious are some glaring errors in content. A fundamental error is that concerning the concept of Shannon's information entropy. To put it bluntly, the concept of entropy has been completely overturned and misinterpreted. If a system (in this case a geographical system) shows an increasing evenness or uniformity in its "information granules", then the entropy is higher (like an expanding gas that uniformly occupies its allowed volume); on the contrary, if the same system moves towards imbalance, asymmetry, disproportion, then the entropy is lower (like a gas in a volume where a concentration gradient is observed). It follows that what is reported in Lines 201-205 is wrong and also Figure 5 is wrong and the calculated values should be the opposite (see my figure in the attached document).

What should be the conceptual or logical link between entropy and environmental quality? Why should a progressive decrease in the level of entropy over time indicate environmental degradation (degeneration and deterioration is the terminology used)?

The four Remote Sensing Ecological Indices (REIs), greenness, wetness, dryness and heat, are simply introduced without any particular explanation. Since they are remotely sensed variables, how can we define them from a physical or radiative point of view, i.e. which spectral bands do they correspond to? How are they measured and expressed? How do they inform the so-called eco-environmental quality? What are the relationships between these variables and the different types of land cover (LCTs)? There are a number of questions that seem to be completely unanswered, and the focus is instead on changes in entropy, where we have no idea what the significance might be.

Figure 2 and Figure 3 show the same things, i.e. they are the same.

There is probably an error in the numerical order of the figures. Figure 10 should be placed after figures 11, 12, 13 and 14.

The "Discussion" section is a slavish repetition of what has already been written in the introductory section and does not add anything new.

Finally, I would like to point out that making an overall assessment of such a vast geographical area is very puzzling, given that there will be profound differences from an environmental point of view between one region and another, which will therefore completely overshadow the special features of the various regions making up ASEAN.

Below you will find some more precise and systematic comments (according to the number of lines). Lack of time prevented me from completing this work. It is, however, an indication of the various things to be corrected in the first part of the manuscript.

[1-4] Alternative title: Measuring the eco-environmental Change in Land Cover in ASEAN countries, from 2001 to 2020, from the Perspective of Knowledge Granularity Entropy

Even a better title choice: Using the concept of Knowledge Granularity Entropy to measure 20-year ecological change of land cover in ASEAN countries.

[7-8] The eco-environment is the basis for the political, economic and social development of any nation or group of nations, such as the Association of Southeast Asian Nations (ASEAN).

[8-11] “There is an inseparable association between land cover change (LCC) and eco-environmental changes (Please, insert a full stop). The spatial and temporal changes in the eco-environment of different land cover types (LCTs) are the driving factors of the regional eco-environment.

Please note: "eco-environmental" and not "eco-environment".

There is also a conceptual error: changes in the eco-environment cannot be "drivers", but "effects", i.e. consequences of some other factors. Please change the sentence accordingly.

Question: Why use the term "eco-environmental" rather than just "ecological"?

[11-14] “This study uses the Remote Sensing Ecological Index (RSEI) to characterize the ecosystem; this issue is solved by breaking down the complex structure of the ecosystem into simple, spatially granular computational units; this approach greatly simplifies the spatial data computation”.

[14-17] 1) “By simplifying and combining the spatial information granules, a model has been proposed based on the concept of 'granularity entropy' of the RSEI values of the different LCTs. This model aims to assess the stability or change of the eco-environment over time”.

[17-19] 2) “The main LCC factors for the decline in eco-environmental quality in ASEAN from 2001 to 2020 are the decrease in forest area and interval changes in eco-environmental index values caused by (???) the increase in grassland area”. Please note: the crossed out part is incomprehensible.

[21] Keywords: “eco-environmental change”

[33] “interdisciplinary field of land systematics”. Better in this form: “interdisciplinary fields of Land System Science”.

Question: Are you sure about the existence of the term “Land Systematics”. It seems a strange and unusual expression to me.

[33-36] “An important factor influencing global ecological changes is represented by the eco-environmental patterns of LCC processes, which should be considered as an important tool for exploring the harmonious coupling of human-land relations and sustainable development pathways”.

[45] “Land ecology has many dynamic feedbacks and processes of change, but it also shows some regularity”.

[51-51] “Existing RSEI studies have mainly focused on assessing the quality of regional Remote Sensing Eco-environment (RSE) over a long period of time”.

Question: What does it mean “quality” of RSE

[52] “land use structure and layout”. Please define / explain this terminology. What do you mean by 'structure' and 'layout'?

[53] Also “ecological stability” is a very critical term that should be explained.

Note: It seems to me that there is a somewhat careless use of terms that have a very precise scientific meaning and should be used with great caution.

[60] The same for the term “volatility”. What does that mean in this context?

[59] “the eco-environmental conditions show a deteriorating trend”

[62-64] “1) each Land Cover Type (LCT) has a different degree of change in RSE over a long time series and will show certain regularity that can be measured”. Please note: the crossed out part should be deleted.

[64-65] “2) There is some relationship between ASEAN land cover dynamics and environmental degradation (as measured by RSE), so we can identify which specific LCTs have caused this decline in environmental quality”.

Note: Again, how do you define environmental quality?

[65-67] “To test these hypotheses, we examined eco-environmental change by means of remote sensing data (from MODIS) of land use from 2001 to 2020 in the ASEAN region”.

[72] The concept of “knowledge granularity entropy” (KGE) should be introduced.

[116-120] “The higher the value of the greenness and wetness indicators and the lower the value of the dryness indicator, the better the eco-environmental conditions. The heat indicator shows an optimum value, while higher or lower values compared to the optimum represent deteriorating environmental conditions”. Note: this rearranged sentence is an attempt to interpret a way of writing that is totally inadequate for understanding.

You wrote [201-205]: “When the RSI gradually tends from a state distribution to a centralized distribution of some indicator information granules, a process of entropy increase of the RSI is formed; in contrast, when the RSI gradually tend from a state distribution to the equilibrium distribution of some indicator information granules, a process of entropy decrease of the RSI is formed”. Apart from the completely shaky English, the concept is wrong and should be reversed. Entropy is a measure of disorder, diversity, evenness. With an equal number granules being considered (16 in the example of Figure 5), when the RSI is characterized by extreme values (very high and very low RSI values) the entropy level is LOW; on the opposite, when the RSI is characterized by evenness values (very similar values one another), then the entropy level is HIGH. Applying the Shannon entropy formula to your example data gives opposite results to those shown in Figure 5.

Author Response

First of all, thank you very much for your questions about my study. These questions are your response based on careful reading of my work. Your misunderstanding mainly comes from the fact that the difference between the knowledge granularity entropy and Shannon information entropy used in this manuscript is not fully understood. Knowledge granularity entropy is also information entropy, and the specific definition can be seen in many literatures, such as

Jianhua Dai, Haowei Tian. Entropy measures and granularity measures for set-valued information systems. Information Sciences, vol.240, pp. 72-82, 2013 .

The formula was originally . The formula in this manuscript is .

Entropy is a measure of disorder, diversity, evenness. Entropy increase is an increase in disorder, while entropy decrease is an increase in order. However, this order or disorder is viewed from the outside of a system, and this study does not discuss the order or disorder of eco-environmental systems. The reason why you think the methods and conclusions of this study were wrong is that your example does not really understand the formula, and your understanding of the pixel value and the number of pixels has a certain deviation. Shannon information entropy is based on the pixel value, your calculation should be image b, and this study is image a. The 5, 7, and 4 in the study represent the number of pixels of spatial information granules 1, 2, and 3, rather than their pixel values.

There are indeed many shortcomings in this study, some of which have been insightful and pointed out sharply by you, which will provide great help for my future research.

I had high expectations for this manuscript; the subject matter immediately struck me as particularly interesting and the approach seemed interesting and innovative. Unfortunately, all I got out of it was a deep disappointment and I am really sad about that. It seems like a wasted opportunity.

In my opinion, there are too many flaws in this manuscript and it should therefore be rejected.

I made several attempts to read it, but I had to make a considerable effort to get to the end and understand its content, its purpose, the methodology used, the results obtained and their relevance.

Firstly, the English language used is often inappropriate for a scientific article. The text is wordy, unnecessarily complicated, redundant in many terms, imprecise, vague, uncertain, sometimes even incorrect. A radical overhaul of the English language would therefore be absolutely necessary, taking into account the need to use correct and specific terminology, to better organize sentences so that they are fully comprehensible, and to avoid too many unnecessary turns of phrase.

Terms are often used without being defined in a rigorous way, as they should appear in a scientific paper. To give you just a few examples, what is meant by 'eco-environmental quality'? The concept of “quality” does not mean anything without a precise definition. Moreover, what is meant by “land use structure and layout”? What is meant by 'structural change'? What does the following expression mean? “The eco-environment shows a deteriorating trend and exhibit some volatility”. What is meant by 'deteriorated or degraded area'? What is meant by “volatility” in this specific context? This is very imprecise and causes confusion. It also increases the frustration of reading. Too often terms are used that should have a precise meaning, but no specific definition is given. This is a far cry from the scientific rigour required. For this reason, it was particularly difficult to read the text and fully understand its meaning.

Response: The terms such as “eco-environmental quality”, “The eco-environment shows a deteriorating trend and exhibit some volatility”. I have conducted research and published on the eco-environmental quality and spatiotemporal changes in ASEAN, and have also cited it in this manuscript. Please refer to the following literature:

[39] Liao, W. Temporal and spatial variations of eco-environment in Association of Southeast Asian Nations from 2000 to 2021 based on information granulation. Journal of Cleaner Production. 2022, 373, 133890.

Even more serious are some glaring errors in content. A fundamental error is that concerning the concept of Shannon's information entropy. To put it bluntly, the concept of entropy has been completely overturned and misinterpreted. If a system (in this case a geographical system) shows an increasing evenness or uniformity in its "information granules", then the entropy is higher (like an expanding gas that uniformly occupies its allowed volume); on the contrary, if the same system moves towards imbalance, asymmetry, disproportion, then the entropy is lower (like a gas in a volume where a concentration gradient is observed). It follows that what is reported in Lines 201-205 is wrong and also Figure 5 is wrong and the calculated values should be the opposite (see my figure in the attached document).

Response: I started writing about this from the beginning, which is the basis for your negation of the manuscript. We can continue to discuss in detail. You are the only one among the five reviewers who pointed out the shortcomings of the manuscript at the methodological level, indicating that your research interests are quite consistent with mine. Regardless of the result of the manuscript, we can further discuss these scientific research issues in the future.

What should be the conceptual or logical link between entropy and environmental quality? Why should a progressive decrease in the level of entropy over time indicate environmental degradation (degeneration and deterioration is the terminology used)?

Response: As you pointed out, I have not yet found a definite mathematical method to solve this problem (the spatial granules association method we are researching may be able to achieve it), and it needs to be achieved through interval value moving by human judgment.  This is also the reason why I did not submit to a higher journal, and I acknowledge this flaw in original manuscript. For more details, see section 5.5 "The entropy increase or decrease phenomenon of the RSE of LCTs is the drastic degree of change of this type of eco environment, and it is necessary to qualitatively determine the evolution direction of eco environment quality by referring to the interval change of RSE indicators”.

The four Remote Sensing Ecological Indices (REIs), greenness, wetness, dryness and heat, are simply introduced without any particular explanation. Since they are remotely sensed variables, how can we define them from a physical or radiative point of view, i.e. which spectral bands do they correspond to? How are they measured and expressed? How do they inform the so-called eco-environmental quality? What are the relationships between these variables and the different types of land cover (LCTs)? There are a number of questions that seem to be completely unanswered, and the focus is instead on changes in entropy, where we have no idea what the significance might be.

Response: At first, I believed that the four indicators of RSEI were widely applied, so they were not defined in the original manuscript. This revised manuscript, combined with your and other reviewers' suggestions, has added definitions for four indicators. Of course, the four indicators representing the eco- environment are not entirely accurate, which is a limitation of quantitative research on the indicator system. No indicator system can fully and accurately represent the research object.

Figure 2 and Figure 3 show the same things, i.e. they are the same.

Response: Because remote sensing indicator data is continuous data, and land cover is classified data. Figure 2 is the spatial information granulation after discretization based on continuous values, while Figure 3 is the spatial information granulation based on classified data.

There is probably an error in the numerical order of the figures. Figure 10 should be placed after figures 11, 12, 13 and 14.

Response: The order in which the reference first appears is the sequence number of the arrangement diagram.

The "Discussion" section is a slavish repetition of what has already been written in the introductory section and does not add anything new.

Response: Perhaps due to differences in writing habits, listing the discussion section separately is because it allows for discussions on innovation, methodology, policies, shortcomings, and other aspects. I adhere to my writing style.

Finally, I would like to point out that making an overall assessment of such a vast geographical area is very puzzling, given that there will be profound differences from an environmental point of view between one region and another, which will therefore completely overshadow the special features of the various regions making up ASEAN.

Response: I have also noticed the question you raised. My published paper “Temporal and spatial variations of eco-environment in Association of Southeast Asian Nations from 2000 to 2021 based on information granulation” proposed an upscaling method for evaluating the eco-environment of ASEAN based on different national policy environments.

Below you will find some more precise and systematic comments (according to the number of lines). Lack of time prevented me from completing this work. It is, however, an indication of the various things to be corrected in the first part of the manuscript.

Response: Thank you very much for your careful reading and the meticulous exploration nature of scientists. I will revise them one by one.

[1-4] Alternative title: Measuring the eco-environmental Change in Land Cover in ASEAN countries, from 2001 to 2020, from the Perspective of Knowledge Granularity Entropy

Even a better title choice: Using the concept of Knowledge Granularity Entropy to measure 20-year ecological change of land cover in ASEAN countries.

Response: My initial title was “Using Knowledge Granularity Entropy to Measure Eco-Environment Change of Land Cover in ASEAN from 2001 to 2020”. I changed the title to the current one when submitting the manuscript. Based on your opinion, I have decided to change the title to “Using Knowledge Granularity Entropy to Measure Eco-environmental Impacts of Land Cover Changes in ASEAN from 2001 to 2020”.

[7-8] The eco-environment is the basis for the political, economic and social development of any nation or group of nations, such as the Association of Southeast Asian Nations (ASEAN).

Response: Thanks, I accept this suggestion.

[8-11] “There is an inseparable association between land cover change (LCC) and eco-environmental changes (Please, insert a full stop). The spatial and temporal changes in the eco-environment of different land cover types (LCTs) are the driving factors of the regional eco-environment.

Please note: "eco-environmental" and not "eco-environment".

There is also a conceptual error: changes in the eco-environment cannot be "drivers", but "effects", i.e. consequences of some other factors. Please change the sentence accordingly.

Response: Thanks, I accept this suggestion.

Question: Why use the term "eco-environmental" rather than just "ecological"?

Response: I believe that ecology refers to the integrity and balance of the environment. Ecology emphasizes the overall nature of the environment, while eco-environment emphasizes the environment itself. So I think “eco-environmental” is more suitable than “ecological”.

[11-14] “This study uses the Remote Sensing Ecological Index (RSEI) to characterize the ecosystem; this issue is solved by breaking down the complex structure of the ecosystem into simple, spatially granular computational units; this approach greatly simplifies the spatial data computation”.

Response: Thanks, I accept this suggestion.

[14-17] 1) “By simplifying and combining the spatial information granules, a model has been proposed based on the concept of 'granularity entropy' of the RSEI values of the different LCTs. This model aims to assess the stability or change of the eco-environment over time”.

Response: Thanks, I accept this suggestion.

[17-19] 2) “The main LCC factors for the decline in eco-environmental quality in ASEAN from 2001 to 2020 are the decrease in forest area and interval changes in eco-environmental index values caused by (???) the increase in grassland area”. Please note: the crossed out part is incomprehensible.

Response: I think this writing is better “The main LCC factors for the decline in eco-environmental quality in ASEAN from 2001 to 2020 are the interval changes in eco-environmental indicator values caused by the decrease in forest area and the increase in grassland area”.

[21] Keywords: “eco-environmental change”

Response: Thanks, I accept this suggestion.

[33] “interdisciplinary field of land systematics”. Better in this form: “interdisciplinary fields of Land System Science”.

Response: Thanks, I accept this suggestion. The original translation is incorrect.

Question: Are you sure about the existence of the term “Land Systematics”. It seems a strange and unusual expression to me.

Response: There is indeed literature mentioning Land System Science, just like the several references cited in manuscript, such as:

Verburg, P. H. ,  Crossman, N. ,  Ellis, E. C. ,  Heinimann, A. ,  Hostert, P. , &  Mertz, O. , et al. (2015). Land system science and sustainable development of the earth system: a global land project perspective. Anthropocene, 12, 29-41.

[33-36] “An important factor influencing global ecological changes is represented by the eco-environmental patterns of LCC processes, which should be considered as an important tool for exploring the harmonious coupling of human-land relations and sustainable development pathways”.

Response: Thanks, I accept this suggestion.

[45] “Land ecology has many dynamic feedbacks and processes of change, but it also shows some regularity”.

Response: Thanks, I accept this suggestion.

[51-51] “Existing RSEI studies have mainly focused on assessing the quality of regional Remote Sensing Eco-environment (RSE) over a long period of time”.

Response: Thanks, I accept this suggestion.

Question: What does it mean “quality” of RSE

Response: Many studies now consider RSEI as a method of evaluating regional eco-environment using these four remote sensing indicators. But I think these four indicators are more reflective of using remote sensing data to assess the eco- environment, and I believe they are remote sensing eco-environment. Based on your suggestion, I think the eco-environment is more suitable than RSE. The “quality” of eco-environment can be found in the cited literature

 [39] Liao, W. Temporal and spatial variations of eco-environment in Association of Southeast Asian Nations from 2000 to 2021 based on information granulation. Journal of Cleaner Production. 2022, 373, 133890.

[52] “land use structure and layout”. Please define / explain this terminology. What do you mean by 'structure' and 'layout'?

Response: Land use structure and layout are very common terms that can be found in many literature and have not been specifically explained. I think the structure is the proportion of their various land cover types, and the layout is the spatial distribution.

[53] Also “ecological stability” is a very critical term that should be explained.

Note: It seems to me that there is a somewhat careless use of terms that have a very precise scientific meaning and should be used with great caution.

[60] The same for the term “volatility”. What does that mean in this context?

Response: Stability is a term widely used in all disciplines, especially regarding the stability of various systems states.

[59] “the eco-environmental conditions show a deteriorating trend”

Response: “Volatility” and “ecological stability” are not the method or conclusions of my research, they are the existing literature on land change and eco-environment change. If we follow this thought, many words in the introduction section need to be defined. In the field of data science, there is a type of data that is metadata. In this way, it may be very diverse or can be defined as meta words that need to be redefined scientifically. Therefore, many published high-level papers need to be rewritten.

Of course, this is my somewhat extreme answer to this question, which may make you angry and deepen your negative views on the manuscript. I agree with your point of view “there is a somewhat careless use of terms that have a very precise scientific meaning and should be used with great caution”. Especially for the method, results, and conclusion sections, many are not clearly defined, so I will try to redefine them as much as possible.

[62-64] “1) each Land Cover Type (LCT) has a different degree of change in RSE over a long time series and will show certain regularity that can be measured”. Please note: the crossed out part should be deleted.

Response: Thanks, I accept this suggestion.

[64-65] “2) There is some relationship between ASEAN land cover dynamics and environmental degradation (as measured by RSE), so we can identify which specific LCTs have caused this decline in environmental quality”.

Note: Again, how do you define environmental quality?

Response: Thanks, I accept this suggestion.

[65-67] “To test these hypotheses, we examined eco-environmental change by means of remote sensing data (from MODIS) of land use from 2001 to 2020 in the ASEAN region”.

Response: Thanks, I accept this suggestion.

[72] The concept of “knowledge granularity entropy” (KGE) should be introduced.

[116-120] “The higher the value of the greenness and wetness indicators and the lower the value of the dryness indicator, the better the eco-environmental conditions. The heat indicator shows an optimum value, while higher or lower values compared to the optimum represent deteriorating environmental conditions”. Note: this rearranged sentence is an attempt to interpret a way of writing that is totally inadequate for understanding.

Response: Thanks, I accept this suggestion.

You wrote [201-205]: “When the RSI gradually tends from a state distribution to a centralized distribution of some indicator information granules, a process of entropy increase of the RSI is formed; in contrast, when the RSI gradually tend from a state distribution to the equilibrium distribution of some indicator information granules, a process of entropy decrease of the RSI is formed”. Apart from the completely shaky English, the concept is wrong and should be reversed. Entropy is a measure of disorder, diversity, evenness. With an equal number granules being considered (16 in the example of Figure 5), when the RSI is characterized by extreme values (very high and very low RSI values) the entropy level is LOW; on the opposite, when the RSI is characterized by evenness values (very similar values one another), then the entropy level is HIGH.

Response: Thank you for your suggestion. Regarding the supplementary definition, the supplementary definition of knowledge granularity is “The knowledge granularity is the measurement for spatial distribution of the spatial information granules. The larger the measurement value, the more evenly distributed the spatial information granules tend to be. ”

The supplementary definition of KGE is “KGE is a measure of the spatial distribution disorder of spatial information granules in an eco-environment system. The larger the KGE value, the higher the disorder. Disorder is not a degradation or evolution of the eco-environment. There is a complementary relationship between KGE and knowledge granularity. The larger the KGE, the smaller the information granularity. The smaller the KGE, the greater the information granularity.”

I provided an explanation for KGE at the beginning. You have made a cognitive mistake. If there is only one value for an indicator, its entropy value is 0, which is the minimum. The calculation of KGE is not related to the height of the indicator value, but only to the number of each spatial information granule.

Due to being a nonnative English speaking country, the English writing level of the manuscript does need to be improved, and I have already contacted AJE company to edit it before submitting.

Applying the Shannon entropy formula to your example data gives opposite results to those shown in Figure 5.

In short, my answer only focuses on the essence of the manuscript itself and the research question, and some may not be polite enough for you. You are the most helpful reviewer among the five in improving the level of manuscript. I firmly believe that I am satisfied with the idea and methods of the study, but my writing skills do need to be improved. Your revisions have greatly benefited me. I believe I have a certain foundation in granular computing, especially in the combination of granular computing and spatial application computing. Your writing level and ability to grasp the structure of scientific research papers make me deeply impressed by your high level of scientific research. If there is an opportunity, we can strengthen cooperation and impact the publication of papers in higher-level journals.

Reviewer 4 Report

This study proposes an entropy increase and entropy decrease model based on the knowledge granularity entropy of remote sensing ecological index (RSEI) for different land cover types (LCTs) in order to assess the stability and change trend of the eco-environment, within a study area.

The study area is the Association of Southeast Asian Nations (ASEAN). Authors considered the spatial and temporal changes in the eco-environment from 2001 to 2020. 

Spatial data calculation were simplified in spatial information granules, the aim was to divide the complex remote sensing eco-environment problem into simpler problems, by using remote sensing ecological index (RSEI), ecological indicators for a complete evaluation of the regional eco-environment. The main aim was to establish which specific land cover types (LCTs) caused the decline of remote sensing ecological quality. The Indicators of the Eco-environment quality aregreenness, wetness, heat and dryness. 

In conclusion, authors stated that the principal causes of LCT changes and environmental decline is during the interval of the 20 years investigated is the significant decrease in evergreen broadleaf forests, increase of woody savannas and climate change.

General comments.

Previous studies on the topic should be compared with results obtained in this study.

Minor revisions:

1) Lines 10-121: Too generic information about formulae used to calculate land indicator.

2) Figure 7: please change number order in the legend

3) Figure 16: improve quality.

4) Line 297: add space

5) Line 417: change title style

Author Response

Thank you for your valuable review comments. Your comments have greatly helped me improve my manuscript and research level. I have made revision to the manuscript item by item based on the opinions of the reviewers.

This study proposes an entropy increase and entropy decrease model based on the knowledge granularity entropy of remote sensing ecological index (RSEI) for different land cover types (LCTs) in order to assess the stability and change trend of the eco-environment, within a study area.

The study area is the Association of Southeast Asian Nations (ASEAN). Authors considered the spatial and temporal changes in the eco-environment from 2001 to 2020. 

Spatial data calculation were simplified in spatial information granules, the aim was to divide the complex remote sensing eco-environment problem into simpler problems, by using remote sensing ecological index (RSEI), ecological indicators for a complete evaluation of the regional eco-environment. The main aim was to establish which specific land cover types (LCTs) caused the decline of remote sensing ecological quality. The Indicators of the Eco-environment quality are greenness, wetness, heat and dryness. 

In conclusion, authors stated that the principal causes of LCT changes and environmental decline is during the interval of the 20 years investigated is the significant decrease in evergreen broadleaf forests, increase of woody savannas and climate change.

General comments.

Previous studies on the topic should be compared with results obtained in this study.

Response: In section 5.3, I have listed a section specifically for comparison and explanation. Compared with RSEI, this method can determine the temporal change direction of each specific land cover type.

Minor revisions:

  • Lines 10-121: Too generic information about formulae used to calculate land indicator.

Response: I initially thought these indicator formulas were very common calculation formulas, but in order to reduce space and repetition rate, I did not include them in the manuscript. Based on the opinions of you and other reviewer experts, I have decided to add the calculation formulas for these indicators.

2) Figure 7: please change number order in the legend

Response: I began to want to use the numbers in Figure 7 to represent the type of land cover, so as to save the space of legend in the image for the following figure. Later, in combination with other review comments, I redraw this figure and some of the following figures.

3) Figure 16: improve quality.

Response: When inserting the output image of Figure 16 and other map results into Word, the resolution of the image will be reduced. If requested by the publishing house, I can provide the original image.

4) Line 297: add space

Response: Thank you for your suggestion. I also think there should be a blank line after the Figure title. I have added a blank line after all the Figure title.

5) Line 417: change title style

Response: Thanks.

Reviewer 5 Report

The paper by Weihua Liao studies the evolution of four remotely sensed parameters (greenness, wetness, dryness, heat) over the territory of ASEAN and their relation to the land cover changes. To perform the computation, the parameter values are discretized (which the author refers to as “information granulation”) and then combined to determine their heterogeneity on the territory through the notion of spatial entropy. Finally, the dynamic of such entropy in time is computed.
The idea is certainly interesting and worth developing, but I think the paper requires some revision.
The title “Measure Eco-environmental Change in Land Cover in ASEAN from 2001 to 2020 from Perspective of Knowledge Granularity Entropy” uses this notion of “eco-environment”, which is only vaguely defined in the paper. I think the paper measures the environmental impacts of land cover changes from THE perspective of knowledge granularity entropy.
This lack of accurate definition characterizes most of the paper, which, though being written correctly from the grammatical viewpoint, sometimes uses terms that do not fit what I assume to be the required meaning. Many sentences need better specification. Examples are “RSI spatial images composed of discrete values according to different values”; “RSI gradually tends from a state distribution to a centralized distribution” (what is a “state distribution” is not defined); “ information granules tend to be concentrated and distributed”; “each indicator is used to calculate the average value of each year by using the Reduce (?) technique”; “any small change in the interval of greenness indicator values will cause larger entropy changes” (larger than what?); “ Permanent snow and ice … tended to cluster around the high interval of heat”; “The eco-environment is the sum (?) of various natural forces” or “moving from the high value interval to the low value interval of indicators” (possibly meaning “from a large interval of values to less dispersed ones”).
Overall, I think the notion of entropy, as used in the paper, should be better related to the classical idea that a larger entropy means more confusion. Here, as shown in fig. 5, if all the land has the same cover type, the value of the entropy is maximum.
Finally, what the paper actually proves is a change in some environmental indicators in relation to the change in land cover types. Whether these changes mean an improvement or a deterioration of the overall environmental quality is not clear from the results since the overall change in entropy is 0.4% in 20 years (according to fig. 8), which appears to be well below the approximation of the data used for the calculation. In quite the same way, all the “Implications for policy-making”, while representing common sense, are not justified by the results presented in the paper.

Minor comments:

-        There are a few typos here and there (e.g., capitalization is random; Fig. 10 is quoted twice meaning other figures; “the RSI gradually tend”; “land coves”; “ASEAN has full of natural resources”).

-        Saying that “All data were calculated according to the calculation formula by using the GEE cloud computing platform (https://code.earthngine.google.com/)” is insufficient for the readers to understand and the link does not work.

-        Similarly, a simple reference to the “formulae for calculating greenness, wetness, dryness and heat” is insufficient since these are the objects of the study, and knowing how they are computed is essential for understanding their accuracy.

-        Fig 3 is the same as fig. 2.

-        Fig. 6 is a bit confusing since the indicators are used one at a time in the following paragraphs

-        The 6% change mentioned in line 256 is better specified later (line 455). Why say it twice with different values?

-        The “Hierarchical change” in 5.2 does not represent a real hierarchy (where the generality or the importance of a level changes). Here, there is simply an addition of other parameters at the same level of importance.

-        Showing the entropy with 9 significant digits (0.652576679) is meaningless, given the approximations (not discussed in the paper) of the data used.

Author Response

Thank you for your valuable review comments. Your comments have greatly helped me improve my manuscript and research level. I have made revision to the manuscript item by item based on the opinions of the reviewers.

The paper by Weihua Liao studies the evolution of four remotely sensed parameters (greenness, wetness, dryness, heat) over the territory of ASEAN and their relation to the land cover changes. To perform the computation, the parameter values are discretized (which the author refers to as “information granulation”) and then combined to determine their heterogeneity on the territory through the notion of spatial entropy. Finally, the dynamic of such entropy in time is computed.
The idea is certainly interesting and worth developing, but I think the paper requires some revision.

The title “Measure Eco-environmental Change in Land Cover in ASEAN from 2001 to 2020 from Perspective of Knowledge Granularity Entropy” uses this notion of “eco-environment”, which is only vaguely defined in the paper. I think the paper measures the environmental impacts of land cover changes from THE perspective of knowledge granularity entropy.

Response: I also think this title is a bit ambiguous. Based on your suggestion, I have decided to change the title to “Using Knowledge Granularity Entropy to Measure Eco-environmental Impacts of Land Cover Changes in ASEAN from 2001 to 2020”, I'm not sure if you think it's suitable.

This lack of accurate definition characterizes most of the paper, which, though being written correctly from the grammatical viewpoint, sometimes uses terms that do not fit what I assume to be the required meaning. Many sentences need better specification. Examples are “RSI spatial images composed of discrete values according to different values”; “RSI gradually tends from a state distribution to a centralized distribution” (what is a “state distribution” is not defined);

Response: I have added a definition explanation for state distribution “State distribution is the organization and location of spatial objects in the objective world, and any object has a state distribution in space”.

 “ information granules tend to be concentrated and distributed”; “each indicator is used to calculate the average value of each year by using the Reduce (?) technique”; “any small change in the interval of greenness indicator values will cause larger entropy changes” (larger than what?); “ Permanent snow and ice … tended to cluster around the high interval of heat”; “The eco-environment is the sum (?) of various natural forces” or “moving from the high value interval to the low value interval of indicators” (possibly meaning “from a large interval of values to less dispersed ones”).

Response: Reduce technique is a common method in GEE, for ease of understanding, I have changed it to the aggregation data technique.

“any small change in the interval of greenness indicator values will cause larger entropy changes” (larger than what?) It would be better to change the word from larger to great. It means that because some land cover types have a small number, the denominator of the entropy formula is small, and small changes in molecules will cause large changes in entropy.

“ Permanent snow and ice … tended to cluster around the high interval of heat”. The original sentence may not be expressed accurately enough, I will change it to “Permanent snow and ice have a trend of moving and clustering from the low value interval of the heat indicator to the high value interval.”.

“The eco-environment is the sum (?) of various natural forces”. I think this sentence is accurate.

“moving from the high value interval to the low value interval of indicators”. The interval does not represent the degree of dispersion, which is represented by the knowledge granularity entropy.

Overall, I think the notion of entropy, as used in the paper, should be better related to the classical idea that a larger entropy means more confusion. Here, as shown in fig. 5, if all the land has the same cover type, the value of the entropy is maximum.

Response: This manuscript does not judge the distribution of land or eco-environmental indicators from the size of entropy, which is of little practical significance. As you mentioned, when each land cover type is the same, the entropy value is the largest. This paper judges the change in eco-environment from the increase and decrease of entropy value of eco-environmental indicators of each land cover type. Of course, there are corresponding quantitative deficiencies, which I also mentioned in section 5.5.

Finally, what the paper actually proves is a change in some environmental indicators in relation to the change in land cover types. Whether these changes mean an improvement or a deterioration of the overall environmental quality is not clear from the results since the overall change in entropy is 0.4% in 20 years (according to fig. 8), which appears to be well below the approximation of the data used for the calculation. In quite the same way, all the “Implications for policy-making”, while representing common sense, are not justified by the results presented in the paper.

Response: I haven't found a mathematical method to quantitatively judge changes in eco-environment quality from entropy changes, and it often relies on human judgment. This is also my current research deficiency in this area, as mentioned in section 5.5 “The entropy increase or decrease phenomenon of the RSE of LCTs is the drastic degree of change of this type of eco-environment, and it is necessary to qualitatively determine the evolution direction of eco-environment quality by referring to the interval change of RSE indicators.” Section “Implications for policy-making” indeed needs improvement. I have made some modifications and hope to improve it in future work and learn more.

Minor comments:

-        There are a few typos here and there (e.g., capitalization is random; Fig. 10 is quoted twice meaning other figures; “the RSI gradually tend”; “land coves”; “ASEAN has full of natural resources”).

Response: Thank you for your careful reading of the manuscript. Your meticulousness is the driving force for me to improve my writing skills. I have carefully checked the manuscript and there may still be some minor errors that are inevitable. The quote in Figure 10 is at the end of each indicator, with a total of five quotes.

-        Saying that “All data were calculated according to the calculation formula by using the GEE cloud computing platform (https://code.earthngine.google.com/)” is insufficient for the readers to understand and the link does not work.

Response: This platform is a remote sensing data cloud computing platform, and it is not a data provider platform. Currently, many researchers around the world are using this platform for remote sensing data calculation and research. Perhaps due to internet restrictions, your computer cannot access this platform and requires the use of scientific internet software.

-        Similarly, a simple reference to the “formulae for calculating greenness, wetness, dryness and heat” is insufficient since these are the objects of the study, and knowing how they are computed is essential for understanding their accuracy.

Response: I initially thought these indicator formulas were very common calculation formulas, but to reduce space and repetition rate, I did not include them in the manuscript. Based on the opinions of you and other reviewer experts, I have decided to add the calculation formulas for these indicators.

-        Fig 3 is the same as fig. 2.

Response: Because remote sensing indicator data is continuous data, and land cover is classified data. Figure 2 is the spatial information granulation after discretization based on continuous values, while Figure 3 is the spatial information granulation based on classified data.

-        Fig. 6 is a bit confusing since the indicators are used one at a time in the following paragraphs

Response: This study analyzes the spatial distribution of changes in the remote sensing eco-environment from each indicator of RSEI and their four combined indicators.

-        The 6% change mentioned in line 256 is better specified later (line 455). Why say it twice with different values?

Response: 6% is the approximate change over the entire 20 years from 2001 to 2020, which is an estimate. The number of changes in line 455 refers to the specific two-year period between 2001 and 2020.

-        The “Hierarchical change” in 5.2 does not represent a real hierarchy (where the generality or the importance of a level changes). Here, there is simply an addition of other parameters at the same level of importance.

 Response: Hierarchy is the prescriptive nature of the system itself, reflecting the research and development process of the system from simple to complex. Remote sensing ecosystem is a complex system with hierarchy in space, time, and attributes. The hierarchy of this manuscript refers more to the hierarchical nature of our approach to ecological problems as different indicators are added. The fewer indicators, the simpler it is to study eco- environmental issues. The more indicators, the more complex it is to study eco- environmental issues, and the more refined and accurate the results.

-        Showing the entropy with 9 significant digits (0.652576679) is meaningless, given the approximations (not discussed in the paper) of the data used.

 Response: Using 9 significant digits does not make much sense, so I have changed the result to 4 significant digits.

Round 2

Reviewer 1 Report

My concerns have been well addressed.

Author Response

Thank you for your recognition of our work. We will continue to work hard to promote research progress in this area.

Reviewer 3 Report

Unfortunately, I have to confirm (this time more briefly, but very sharply) my negative evaluation of this manuscript.

The most important objection concerns the concept of "entropy", which is one of the scientific pillars of mankind. No misleading interpretation can affect the basis of this fundamental thermodynamic concept. If, as stated at the beginning of Section 3.3 (lines 209-210), entropy, or better "information entropy", is a measure of disorder and a degree of evenness, then whatever index one devises as an "entropy" indicator should be consistent with this basic definition, otherwise an unacceptable conceptual "overturning" is made.

I will mention it again (and for the last time): a system moving towards greater disorder develops a high level of entropy, i.e. a state of greater balance in its components, greater evenness, homogeneity, no gradient, low free energy. Exactly the opposite is observed when a system evolves towards a low entropy state, in which case order prevails, i.e. a state of denseness, closeness, higher heterogeneity in its components, steep concentration gradients, etc.

No matter what the application, no matter what the formula, this concept cannot be overturned. At best it is the other way round. It is you who must adapt to this basic concept and apply it in the right way.

Lines 243-247: “When the RSI gradually tends (from a state distribution) to a centralized distribution of some indicator information granules, a process of entropy increase of the RSI is formed; in contrast, when the RSI gradually tends (from a state distribution) to the equilibrium distribution of some indicator information granules, a process of entropy decrease of the RSI is formed”.

 Lines 252-253: “the three levels of greenness indicators tend to be more balanced in spatial distribution, which is an entropy decreasing process”.

 Lines 255-257: “The three levels of greenness indicators tend to be spatially concentrated in level 2, and the greenness spatial information granules tend to be concentrated and distributed, which is an entropy increasing process”.

No, these sentences are not correct and upset the foundations of science. Have I made myself clear enough? If you want to apply the entropy concept to “knowledge granularity”, you are absolutely welcome, but please but make sure that this application is in the right direction and not the other way around.

I did not receive any replies about the use of vague and undefined terms such as eco-environmental 'quality', 'deteriorating trend', 'volatility', ‘land use structure and layout’, etc. I assume that these qualitative terms may not be considered appropriate enough in a scientific paper without a clear definition. And unfortunately, definitions are still missing.

When I ask you to justify the conceptual or logical link between entropy and environmental quality, your answer is disarming: "I have not yet found a definite mathematical method to solve this problem". Therefore, all the figures you have presented in the manuscript are just a useless mathematical exercise! You did not explain this in the manuscript!

The function: RSEI = f(greenness; wetness; dryness; heat) is still missing. What are the values and signs of the weighting coefficients that should be included in the formula? How were these weighting factors determined?

I asked you why to use the term "eco-environmental" rather than just "ecological"? You replied that “ecology refers to the integrity and balance of the environment. Ecology emphasizes the overall nature of the environment, while eco-environment emphasizes the environment itself”. Therefore you concluded that “eco-environmental” is more suitable than “ecological”.

In my turn, I have to stress that “ecology” refers to the integration of both geo-physical-climatic (environmental) and biological features; indeed it is considered as the integration of “biotope” (the physical milieu) and “biocenosis” (the living community). Therefore “ecological” should be the comprehensive term to be used. It was originally proposed in this way by

I still think that the English of the text could be greatly improved, especially with regard to a more economical and careful use of technical terms. Some terms have precise scientific meanings and should not be used lightly. Most of my language suggestions were accepted, but they were just a few examples limited to the first part of the manuscript. In my opinion, the problem still exists.

In general, I remain convinced that the manuscript is not suitable for publication. It remains the Editor's decision whether or not to publish it.

Author Response

First of all, thank you again for your valuable comments. I have carefully considered your comments and made detailed modifications. The main point of disagreement lies in the concept of “entropy”. The paper uses knowledge granularity entropy, which, along with the knowledge granularity in the paper, is a formula for measuring information uncertainty based on knowledge partitioning. These entropy formulas based on the granular computing concept mainly used in this article have been proven in many literature, such as:

https://doi.org/10.1016/j.asoc.2009.03.007

https://doi.org/10.1016/j.ins.2013.03.045

In this very famous Chinese journal (Science in China Series F-Information Sciences (in Chinese)), the author proposed rough entropy and has a theorem as shown in the following list figure, here is the total number of samples.

https://doi.org/10.1360/zf2008-38-12-2048

From the formula, it can be seen that the entropy of these measures based on granular computing is complementary to Shannon entropy, meaning that when measuring uncertainty, the measurement values will be opposite to Shannon entropy values. Based on the above theory, I have carefully considered your comments and deeply realized that there were some inappropriate and imprecise aspects in my previous manuscript. The concept of "entropy increasing and decreasing" mentioned earlier is only an increase or decrease in the KGE value, not a change in the degree of chaos in the system. It is just a movement of the indicator value range, which is the change between the spatial information granules in the index.

Unfortunately, I have to confirm (this time more briefly, but very sharply) my negative evaluation of this manuscript.

The most important objection concerns the concept of "entropy", which is one of the scientific pillars of mankind. No misleading interpretation can affect the basis of this fundamental thermodynamic concept. If, as stated at the beginning of Section 3.3 (lines 209-210), entropy, or better "information entropy", is a measure of disorder and a degree of evenness, then whatever index one devises as an "entropy" indicator should be consistent with this basic definition, otherwise an unacceptable conceptual "overturning" is made.

Response: Knowledge granularity and knowledge granularity entropy is actually an average measure of the different levels of information refinement. Our research has made some of the mistakes you mentioned and made corresponding modifications. This paper mainly uses the change of knowledge granularity entropy value combined with the movement of spatial information granules in remote sensing indicators to determine the interval movement of eco-environment indicators. Research does not focus on the stability or changes in the stability of eco-environmental systems

I will mention it again (and for the last time): a system moving towards greater disorder develops a high level of entropy, i.e. a state of greater balance in its components, greater evenness, homogeneity, no gradient, low free energy. Exactly the opposite is observed when a system evolves towards a low entropy state, in which case order prevails, i.e. a state of denseness, closeness, higher heterogeneity in its components, steep concentration gradients, etc.

No matter what the application, no matter what the formula, this concept cannot be overturned. At best it is the other way round. It is you who must adapt to this basic concept and apply it in the right way.

Response: As mentioned in the initial answer, the knowledge granularity entropy we use has been proven correct and exists in many papers. Our mistake is to describe it as a measure of the degree of system disorder. In fact, we did not use knowledge granularity entropy to assess the degree of disorder in the eco-environmental system. It is only a measure of the degree of problem refinement, and we combine the interval movement of spatial information granules to determine the trend of changes in eco-environment indicators.

Lines 243-247: “When the RSI gradually tends (from a state distribution) to a centralized distribution of some indicator information granules, a process of entropy increase of the RSI is formed; in contrast, when the RSI gradually tends (from a state distribution) to the equilibrium distribution of some indicator information granules, a process of entropy decrease of the RSI is formed”.

 Lines 252-253: “the three levels of greenness indicators tend to be more balanced in spatial distribution, which is an entropy decreasing process”.

 Lines 255-257: “The three levels of greenness indicators tend to be spatially concentrated in level 2, and the greenness spatial information granules tend to be concentrated and distributed, which is an entropy increasing process”.

Response: This is the most significant mistake we have made, and it is also your constant criticism of our incorrect research. We accept this and sincerely thank you. The process of entropy increase is a spontaneous progression from order to disorder, and entropy decrease is the inverse of entropy increase. Our research is not about studying the order and disorder of the eco-environment. We only studied the interval movement of spatial information granules in eco-environment indicators, but mistakenly used these two concepts. The previously used entropy increase is a spatial aggregation process of indicator granules, while entropy decrease is a spatial dispersion process of indicator granules. Based on these, we have made corresponding modifications to the entire manuscript.

No, these sentences are not correct and upset the foundations of science. Have I made myself clear enough? If you want to apply the entropy concept to “knowledge granularity”, you are absolutely welcome, but please but make sure that this application is in the right direction and not the other way around.

I did not receive any replies about the use of vague and undefined terms such as eco-environmental 'quality', 'deteriorating trend', 'volatility', ‘land use structure and layout’, etc. I assume that these qualitative terms may not be considered appropriate enough in a scientific paper without a clear definition. And unfortunately, definitions are still missing.

Response: We still don't agree with this comment, even if you reject my revised manuscript again. We acknowledge that the terms in the Materials and data, Methods, Results, Discussion and Conclusion section need to be defined, but many of the terms you mentioned are in the Introduction section, and we have also marked the references. And everyone's understanding of term may vary, so this kind of modification will be endless. Of course, we can answer these terms here.

eco-environmental quality: For many existing RSEI evaluation studies, the larger the calculated value, the better the eco-environmental quality. If we understand the eco-environmental quality from a single indicator, the larger the values of greenness and wetness, the greater the contribution to the eco-environmental quality. The larger the values of dryness, the smaller the contribution to the eco-environmental quality. The heat indicator is a moderate indicator that contributes less and less to the eco-environmental quality from a certain value in the middle (the average value taken in this study) to both sides.

Although we have mentioned eco-environment quality multiple times in the manuscript, we do not consider it to be a particularly explanatory term. Moreover, we have already explained the relationship between indicators and eco-environmental conditions in the section 3.1. Indicators of the Eco-environment in our first submission.

deteriorating trend and volatility: This is the conclusion of our published research, as shown in the following figure. It can be seen that the long-term calculation of eco-environment quality results shows a gradually decreasing trend, which is deteriorating trend. From the calculation results, we can also see that not every year is smaller than the previous year's result value, and the results show a certain degree of volatility.

land use structure and layout: Land use structure is a proportional relationship between various land types in terms of quantity, while land use layout is a mutual relationship between various land types in terms of spatial distribution. Changes in land use structure and layout refers to changes in land types in terms of quantity, spatial relationships, or both.

When I ask you to justify the conceptual or logical link between entropy and environmental quality, your answer is disarming: "I have not yet found a definite mathematical method to solve this problem". Therefore, all the figures you have presented in the manuscript are just a useless mathematical exercise! You did not explain this in the manuscript!

Response: The formula (9) in the revised manuscript can determine whether the spatial information of eco-environment indicators in time series is spatially aggregated or spatially dispersed, which cannot be concluded by comparing the movement of indicators alone. We only rely on human judgment to determine whether the spatial information granules in the indicator are moving between high value intervals or low value intervals, which is also our next direction of effort.

The function: RSEI = f(greenness; wetness; dryness; heat) is still missing. What are the values and signs of the weighting coefficients that should be included in the formula? How were these weighting factors determined?

Response: There is a misconception here that we are studying the changes in knowledge granularity entropy for each indicator and combination of indicators, not the changes in knowledge granularity entropy for the final RSEI evaluation results. Many studies have been conducted on RSEI evaluation, and our research is not related to RSEI evaluation. That means we only used the indicators from RSEI. Therefore, we have provided a definition for each indicator in the revised manuscript and do not need to provide a mapping function f.

Although we have a section Comparison with RSEI. But this is based on the comparison of existing research results and methods, and the weights of RSEI indicators vary from year to year. As we have always emphasized, all the content of this study is based on RSEI indicators or combination indicators, and is not related to the RSEI evaluation results.

I asked you why to use the term "eco-environmental" rather than just "ecological"? You replied that “ecology refers to the integrity and balance of the environment. Ecology emphasizes the overall nature of the environment, while eco-environment emphasizes the environment itself”. Therefore you concluded that “eco-environmental” is more suitable than “ecological”.

In my turn, I have to stress that “ecology” refers to the integration of both geo-physical-climatic (environmental) and biological features; indeed it is considered as the integration of “biotope” (the physical milieu) and “biocenosis” (the living community). Therefore “ecological” should be the comprehensive term to be used. It was originally proposed in this way by

Response: Thank you for your guidance and sincere response. We have deepened our understanding of ecology and the eco-environment.

I still think that the English of the text could be greatly improved, especially with regard to a more economical and careful use of technical terms. Some terms have precise scientific meanings and should not be used lightly. Most of my language suggestions were accepted, but they were just a few examples limited to the first part of the manuscript. In my opinion, the problem still exists.

Response: As stated in the first round of responses, as an author from a nonnative English speaking country, there may be some shortcomings in English writing. We have submitted a translation company and edited the entire manuscript. Although our writing habits may not meet your requirements in many aspects, we have made every effort and the other four reviewers highly agree with the English writing of this manuscript.

In general, I remain convinced that the manuscript is not suitable for publication. It remains the Editor's decision whether or not to publish it.

Response: The process of revising a manuscript is a process of continuous improvement and learning. Through your discussion, we have gained a lot, which is reflected in various aspects such as rigor, writing habits, and research objectives. Regardless of the final result we sincerely appreciate your valuable comments.

Finally, based on your understanding of entropy, I would like to ask a question unrelated to this study but related to entropy.

Regarding spatial distribution, the entropy values in the following two cases are the same regardless of which formula is used to measure them, but their spatial distribution and stability are actually different. We still cannot find a formula to distinguish their variables such as disorder and stability.

Round 3

Reviewer 3 Report

Thank you for your further clarifications. 

All in all, I remain convinced of the inadequacy of your work, which I respect in any case, and I also appreciate the author for his considerable efforts.

I do not consider it appropriate to proceed further with error reporting on my part and consequent corrections on your part (we have now reached the third round). I have therefore referred the matter back to the Editor.

Author Response

Thank you for your partial recognition of our work. We will continue to work hard to promote research progress in this area. Based on your further feedback, we have made English revisions to the manuscript sentence by sentence.